# AVERE: Improving Audiovisual Emotion Reasoning with Preference Optimization

**Ashutosh Chaubey, Jiacheng Pang, Maksim Siniukov & Mohammad Soleymani**
Institute for Creative Technologies
University of Southern California
Los Angeles, CA 90007, USA
`achaubey@usc.edu` & `soleymani@ict.usc.edu`

## Abstract

Emotion understanding is essential for building socially intelligent agents. Although recent multimodal large language models (MLLMs) have shown strong performance on this task, two key challenges remain: (i) spurious associations between emotions and irrelevant audiovisual cues (*reasoning errors*) and (ii) hallucination of audiovisual cues (*perception errors*) driven by text priors in the language model backbone. To quantify and understand these issues, we introduce **EmoRe-AlM**, a benchmark designed to evaluate MLLMs for cue–emotion associations, hallucinations and modality agreement. We then propose **AVEm-DPO**, a preference optimization technique that aligns model responses with both audiovisual inputs and emotion-centric queries. Specifically, we construct preferences over (i) responses exhibiting spurious associations or hallucinations and (ii) audiovisual input pairs guided by textual prompts. We also include a regularization term that penalizes reliance on text priors, thereby mitigating modality-specific cue hallucinations. Experimental results on DFEW, RAVDESS and EMER demonstrate that our method significantly improves the performance of the reference baseline models (6-19% of relative performance) in zero-shot settings. By providing both a rigorous benchmark and a robust optimization framework, this work enables principled evaluation and improvement of MLLMs for emotion understanding and social AI. Code, models and benchmark at our project page - avere-iclr.github.io.

## 1 Introduction

Emotion understanding is essential for social AI agents to generate tailored responses and foster meaningful human–machine interactions (Chaturvedi et al., 2023; Kolomaznik et al., 2024; Elyoseph et al., 2024). Emotion perception also finds applications in domains such as health (Balcombe & De Leo, 2022; Litendahl et al., 2025) and education (Salloum et al., 2025), where appropriately responding to affective states can improve therapeutic alliance and learning outcomes.

Traditional multimodal emotion recognition methods (Sun et al., 2023; Wang et al., 2023; Chen et al., 2024) lack interpretability, as they only perform classification without grounding responses in audiovisual cues. Moreover, emotion is a complex and multi-componential construct that extends beyond the basic emotion labels that can be assigned by supervised learning methods (Ekman & Friesen, 1978; Scherer, 2005). To address these challenges, recent approaches leverage multimodal large language models (MLLMs) to generate detailed emotion descriptions for interpretability (Cheng et al., 2024; Huang et al., 2025a) and to output emotion-related keywords that cover a broader spectrum of emotional states (Lian et al., 2024; 2025a).

However, audiovisual MLLMs are susceptible to *hallucinations*, frequently generating inaccurate or fabricated responses (Li et al., 2023; Sahoo et al., 2024). In the context of emotion understanding, they face two critical bottlenecks, as illustrated in Fig. 1. First, these models often ground emotion predictions on irrelevant cues (e.g., attire color, ambient noise) – *reasoning errors*. Second, they tend to hallucinate additional cues in their responses to justify emotions – *perception errors*. Such hallucinations are largely driven by text priors in the language model backbone, which bias the model to include cues that commonly co-occur with specific emotions (e.g., associating tears

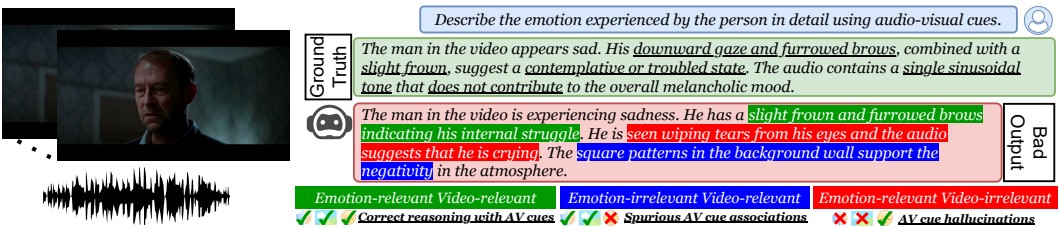

Figure 1: Existing MLLMs (i) include spurious associations between AV cues and emotions – *reasoning errors* (*blue highlight*) and (ii) hallucinate AV cues to explain emotions – *perception errors* (*red highlight*). AV: audiovisual.

with the sound of crying). The scarcity of high-quality, emotion-specific instruction tuning datasets (Cheng et al., 2024; Lian et al., 2025a) further aggravates these issues. Addressing these challenges is essential, as they compromise the reliability of MLLM agents in social interactions and complex emotion reasoning scenarios.

Existing emotion reasoning benchmarks (Lian et al., 2023b; 2024) lack the diverse and complex samples needed to fully evaluate these issues. Additionally, current audiovisual hallucination benchmarks (Sung-Bin et al., 2025; Leng et al., 2025) predominantly focus on object-level hallucinations in audio or video, rather than on emotion-specific reasoning. Moreover, many existing MLLMs (Cheng et al., 2024; Lian et al., 2025a) rely on two-stage evaluation pipelines involving an external (often proprietary) LLM such as GPT (OpenAI et al., 2024), making replication and benchmarking difficult. To address these limitations, we introduce the **EmoReAlM** benchmark, a comprehensive suite of multiple-choice question–answer (MCQA) tasks designed to evaluate audiovisual emotion reasoning, modality agreement and hallucination-related stress tests (Fig. 2). The MCQA format enables transparent, reproducible and scalable evaluation of MLLMs on emotion-centric tasks without requiring additional LLMs during inference.

Evaluation of recent MLLMs on our benchmark highlights spurious association and hallucination issues outlined in Fig. 1. To address these limitations, we propose **AVEm-DPO** – a multimodal direct preference optimization (DPO) technique (Rafailov et al., 2023) to enhance the emotion reasoning capabilities of MLLMs. In particular, we design explicit prompt-based audiovisual input preferences to mitigate hallucinations caused by cross-modal interactions. We also introduce text-prior debiasing, which penalizes policy reward for responses to text-only inputs. Together, these techniques significantly improve the performance of reference MLLMs, outperforming all baselines in zero-shot evaluation on both our benchmark and existing emotion recognition and reasoning datasets.

To summarize, the main contributions of our work are:

- We introduce the **EmoReAlM** benchmark with **4000 human-verified** MCQA samples to evaluate emotion reasoning and emotion-related hallucinations in MLLMs, highlighting bottlenecks such as spurious audiovisual cue associations and hallucinated cues for explaining emotions.
- We propose **AVEm-DPO**, a direct preference optimization technique that enforces explicit prompt-based modality preferences and reduces text-only model biases through a regularizer that penalizes over-reliance on text priors.
- We conduct extensive evaluations of existing MLLMs, demonstrating current bottlenecks and showing the superior performance of the proposed DPO-trained models in zero-shot settings.

## 2 RELATED WORK

**MLLMs for Emotion.** While general MLLMs (Zhang et al., 2024; Lin et al., 2024; Zhang et al., 2025a; Xu et al., 2025b; Li & team, 2025) show non-trivial emotion recognition ability (Cheng et al., 2024), several studies pursue domain-specific instruction tuning (Xie et al., 2024; Chaubey et al., 2025; Yang et al., 2025). EmotionLLaMA(Cheng et al., 2024) is an audiovisual LLM for emotion recognition and captioning, finetuned on a limited dataset (≈30k samples). Lian et al. (2024) introduces open-vocabulary emotion recognition (OV-ER), and AffectGPT (Lian et al., 2025a) employs a lightweight audiovisual fusion projector for OV-ER. EmotionQwen (Huang et al., 2025a) im-

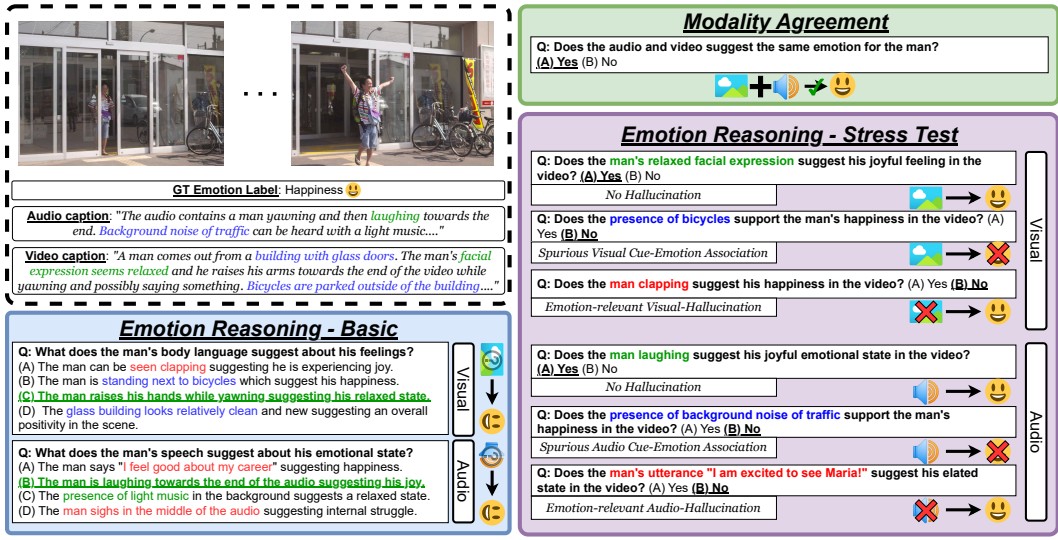

Figure 2: **EmoReAlM Tasks**. In addition to basic emotion reasoning, we include tasks for *Modality Agreement* and *Emotion Reasoning - Stress Test* to test spurious cue-emotion associations and cue hallucinations. Red text is a hallucinated cue, blue text is an emotion-irrelevant cue and green text is a cue relevant for emotion understanding. Correct choices are **underlined**.

proves emotion understanding while preserving general skills via a mixture-of-experts router. Han et al. (2025b) use modality-specific experts with attention reallocation to handle audiovisual emotion mismatch, and Wen et al. (2025) leverage retrieval-augmented generation with chain-of-thought for better reasoning. In contrast, we improve reasoning through multimodal preference optimization and text-prior debiasing.

Rigorous evaluation of multimodal emotion reasoning requires diverse, systematic benchmarks. Lian et al. (2023b) provide detailed descriptions of transcript, audio and visual cues for emotion reasoning, which can support GPT-based evaluation (Cheng et al., 2024; Han et al., 2025b). Xing et al. (2025) present a holistic benchmark spanning text, image, video and audio hallucinations related to emotions. Our benchmark instead focuses squarely on audiovisual emotion understanding with a standardized pipeline and tasks beyond hallucination, including modality agreement and spurious cue–emotion associations.

**Preference Optimization.** Direct preference optimization (DPO) (Rafailov et al., 2023; Liu et al., 2025a) was introduced to align LLMs to human preferences. DPO has also emerged as a leading approach for mitigating hallucinations in vision LLMs (Yu et al., 2024; Wang et al., 2024; Sarkar et al., 2025; Huang et al., 2025b; Liu et al., 2025b; Zhang et al., 2025b), but its use in audiovisual LLMs remains limited. VistaDPO (Huang et al., 2025b) increases video LLM robustness by building instance-level, temporal-level and object-level preferences of video inputs. Sun et al. (2025) apply process DPO for step-wise audiovisual reasoning, while Tang et al. (2025) use multi-round DPO for audiovisual captioning. Luo et al. (2025) employ DPO for emotional speech alignment to improve Omni-LLM outputs. Ye et al. (2025) construct multimodal preference data via ambiguity scoring, and Lian (2025) use group relative policy optimization for AffectGPT. Concurrently, Omni-DPO (Chen et al., 2025) studies audiovisual modality preference. Our method differs by constructing prompt-based audiovisual preference pairs for fine-grained alignment and by introducing text-prior debiasing to reduce hallucinations in MLLMs.

## 3 EMOREALM BENCHMARK

Fig. 2 shows different tasks present in the proposed **EmoReAlM** Benchmark. The goal of this benchmark is to test the reasoning capabilities of MLLMs to judge the *emotion experienced by the character in the given video*, specifically over the following verticals – (i) **reasoning the correct emotion** with relevant audiovisual cues (ii) identifying whether the inferred emotion from **audio and video are in agreement** (iii) testing the **association of perceived audiovisual cues** with different

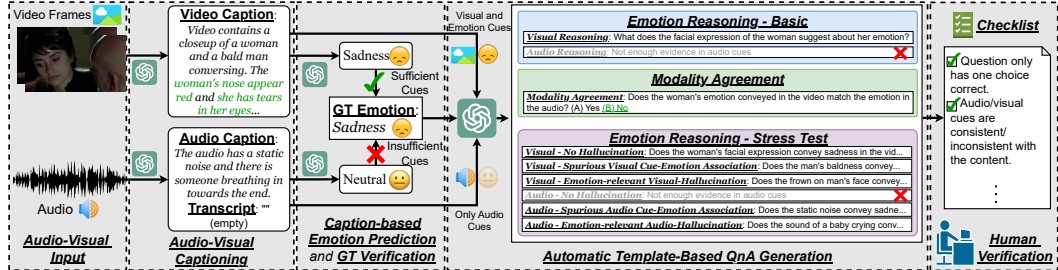

Figure 3: **EmoReAlM Creation Pipeline**. We first disentangle the audiovisual information by separate captioning and verify the cues with text-based emotion prediction to find emotion-relevant cues. Finally, GPT-4o is used to generate MCQA samples that are later verified manually.

emotions (*reasoning errors*) and (iv) testing **audiovisual hallucination due to text-only emotion-related biases** (*perception errors*).

## 3.1 TASK DESCRIPTIONS

**Emotion Reasoning – Basic.** This task evaluates an MLLM's ability to identify and reason about the emotion experienced by a person in a video by linking appropriate audio (e.g., speech transcription, tone) and visual (e.g., facial expression, body language) cues to specific emotions. To increase difficulty, the ground-truth emotion is not provided in the question. Incorrect options are constructed by modifying the correct answer to include either emotion-irrelevant cues present in the video or hallucinated cues that falsely justify the emotion.

**Modality Agreement.** This task assesses whether the audio and visual modalities convey the same emotional state. Unlike AVHBench (Sung-Bin et al., 2025), which focuses on general cross-modal alignment, this task specifically targets agreement in emotional interpretation across modalities.

**Emotion Reasoning – Stress Test.** MLLMs are vulnerable to both *reasoning errors* and *perception errors*: the former lead the model to base its responses on irrelevant audiovisual cues present in the input, while the latter cause it to rely on hallucinated cues that are not actually present. This task probes MLLMs for susceptibility to spurious cue-emotion associations (*perception errors*) and hallucinated explanations driven by language model biases (*reasoning errors*). Each question follows the format: *"Does the {audio/visual cue} suggest {emotion} of the character?"*. For a modality X, we define three sub-tasks: (i) No Hallucination — correctly associating an audio/visual cue with the appropriate emotion. (ii) Spurious X Cue-Emotion Association — linking emotion-irrelevant cues to the correct emotion. (iii) Emotion-Relevant X-Hallucination — associating the correct emotion with a hallucinated cue that typically co-occurs with it. For example, in Fig. 2, a man is not clapping (per the visual caption), yet a hallucination-based question associates clapping with happiness—since clapping is commonly linked to positive emotions like joy.

## 3.2 AUTOMATIC DATA CREATION

Fig. 3 shows the automatic pipeline used to construct the *EmoReAlM* benchmark. Our approach builds on existing manually labeled audiovisual emotion recognition datasets that provide single-word emotion annotations. For each video, we first use an MLLM to extract detailed audio and visual captions separately, effectively disentangling the two modalities. These captions describe both emotion-relevant and irrelevant cues. To verify whether either modality reflects an emotion, we prompt an LLM to classify the audio and video captions independently into one of seven categories of neutral, in addition to six basic emotions Ekman (2005). Samples are discarded if neither caption yields a valid emotion label. Given the validated captions and emotion label, we then generate tailored prompts and question templates for each task described in Section 3.1. This modality-wise captioning and emotion verification process ensures the construction of high-quality, verifiable MCQA pairs that reflect meaningful audiovisual cue associations. More details and prompts are present in Appendix B.

**Details.** All videos are sourced from the DFEW dataset (Jiang et al., 2020). GPT-4o (OpenAI et al., 2024) is used for caption extraction, emotion classification and question–answer pair generation.

### 3.3 POST-PROCESSING AND HUMAN VERIFICATION

We employ GPT-4o (OpenAI et al., 2024), Gemini-2.5 (Gemini-Team et al., 2025) and Qwen-2.5 (Qwen-Team et al., 2025) to predict the correct answer to the generated questions just by using question text as input. We remove all the QA pairs for which all the models identified the correct answer just with the text information. Finally, since the QA samples are generated automatically leveraging MLLMs, which can hallucinate themselves, we perform a human verification over the samples generated by recruiting over 470 participants using the crowd-sourcing platform Prolific. Details are present in Appendix B.2.

### 3.4 BENCHMARK STATISTICS

Table 1 summarizes the data statistics of the proposed *EmoReAlM* Benchmark, which comprises a total of **4,000 questions** over **2,649 unique videos**. Samples from the benchmark are present in Appendix B.5. Importantly, for tasks which always have a fixed set of answer choices (*Emotion Reasoning - Stress Test* and *Modality Agreement – Yes/No*), we ensure that there is a uniform distribution of correct answer texts over the possible answer choice texts. Additionally, we ensure that the distribution of emotion labels over the videos in the benchmark matches the video source dataset (refer to Appendix B.3 for details). It is also important to note that *EmoReAlM* is only used as a **test set** to evaluate the reasoning capabilities of MLLMs, and we use a different dataset for preference optimization (refer Section 4.3).

Table 1: *EmoReAlM* Benchmark Statistics.

| Task | | # QA | # vid. | Rand. Acc. |
|---|---|---|---|---|
| Reasoning Basic | Audio | 972 | 784 | 25% |
| | Visual | 1024 | 883 | 25% |
| Modality Agreement | | 456 | 456 | 50% |
| Reas. Stress Test | Audio | 820 | 655 | 50% |
| | Visual | 728 | 593 | 50% |
| **Total** | | **4000** | **2649** | |

## 4 AVEM-DPO

Direct preference optimization (DPO) (Rafailov et al., 2023) aligns LLMs to human preferences, bypassing the need to develop a reward model. In the context of audiovisual LLMs, given a reference model $\pi_{\text{ref}}$, we can reformulate the DPO objective to learn an optimal policy $\pi_\theta$ as the following,

$$\max_{\pi_\theta} \mathbb{E}_{(a,v,x)\sim\mathcal{D}, y\sim\pi_\theta(\cdot|a,v,x)} \left[ r(a,v,x,y) \right] - \beta \mathbb{D}_{\text{KL}}(\pi_\theta(\cdot \mid a,v,x) \parallel \pi_{\text{ref}}(\cdot \mid a,v,x)) \tag{1}$$

where $(a, v)$ is audiovisual input, $x$ is text prompt, $y$ is text response and $r(a, v, x, y)$ is the reward function for given input-output pair. Optimizing Eq. (1) to find optimal policy results in the following reward formulation,

$$r(a,v,x,y) = \beta \log \frac{\pi_\theta(y \mid a,v,x)}{\pi_{\text{ref}}(y \mid a,v,x)} + \beta \log Z(a,v,x) \tag{2}$$

where $Z(\cdot)$ is the partition function derived in Rafailov et al. (2023). With access to a preference dataset $\mathcal{D}_y^{\text{pref}}$ with samples $(a, v, x, y_w, y_l)$ and using the Bradley-Terry preference model (Bradley & Terry, 1952) to model preference of chosen response $(y_w)$ over rejected response $(y_l)$, the final DPO objective becomes

$$\mathcal{L}_{\text{DPO}} = -\mathbb{E}_{(a,v,x,y_w,y_l)\sim\mathcal{D}^{\text{pref}}} \left[ \log \sigma \left( \beta \log \frac{\pi_\theta(y_w \mid a,v,x)}{\pi_{\text{ref}}(y_w \mid a,v,x)} - \beta \log \frac{\pi_\theta(y_l \mid a,v,x)}{\pi_{\text{ref}}(y_l \mid a,v,x)} \right) \right] \tag{3}$$

### 4.1 MULTIMODAL PREFERENCE OPTIMIZATION

Naive DPO (Eq. (3)) applied to MLLMs, when relying only on response preference, often causes the policy model to overfit to the input prompt $x$ while neglecting the multimodal inputs during alignment (Wang et al., 2024; Sarkar et al., 2025). To address this limitation, preference optimization can be extended to incorporate audiovisual inputs as follows:

$$\mathcal{L}_{\text{DPO}}^{av} = -\mathbb{E}\left[ \log \sigma(u(a_w, v_w, a_l, v_l, x, y_w)) \right], \ u(\cdot) = \beta \log \frac{\pi_\theta(y_w \mid a_w, v_w, x)}{\pi_{\text{ref}}(y_w \mid a_w, v_w, x)} - \beta \log \frac{\pi_\theta(y_w \mid a_l, v_l, x)}{\pi_{\text{ref}}(y_w \mid a_l, v_l, x)} \tag{4}$$

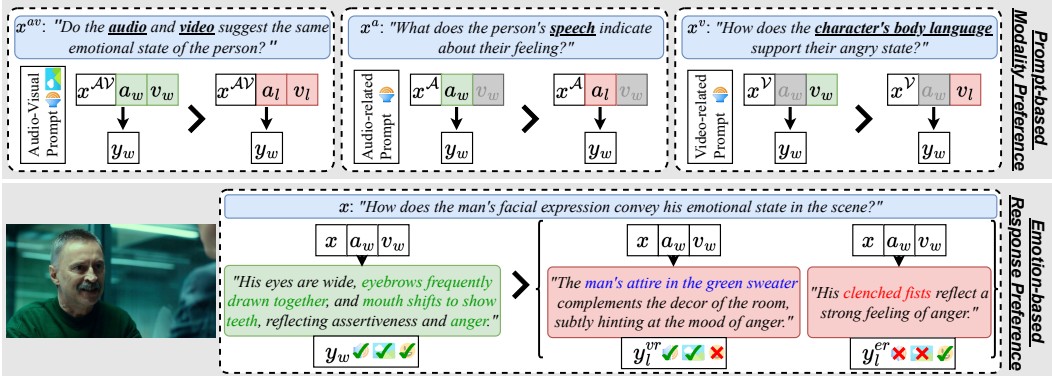

Figure 4: **Preference pairs in AVEm-DPO**. *(Top)* Fine-grained preference over modality input based on current prompt. *(Bottom)* Each chosen response $y_w$ has two rejected responses — $y_l^{vr}$ relevant to the video but with spurious emotion association and $y_l^{er}$ irrelevant to the video (hallucinated) but related to the emotion.

where $(a_w, v_w)$ and $(a_l, v_l)$ denote the chosen and rejected multimodal inputs. This objective ensures that the policy model aligns its response $y_w$ to the correct (chosen) audiovisual input $(a_w, v_w)$.

**Prompt-based Modality Preference (PMP).** While Eq. (4) enforces preference over $non\text{-}text$ inputs, in the case of audiovisual (or *"omni"*) LLMs the input prompt $x^m$ may relate to both audio and visual modalities, or to only one of them ($m \in \mathcal{M} = \{\mathcal{AV}, \mathcal{A}, \mathcal{V}\}$). This often leads to cross-modality-induced hallucinations in MLLMs (Sung-Bin et al., 2025), where a response to a prompt concerning one modality $x^{m_1}$ is spuriously influenced by another modality $m_2 \in \mathcal{M} \setminus \{m_1\}$.

To mitigate this issue, we construct the preference dataset $\mathcal{D}_{av}^{\text{pref}}$ with fine-grained modality-level preferences conditioned on the input prompt $x^m$, as illustrated in Fig. 4 *(Top)*. For example, for a query specific to one modality $x^m$ (e.g., visual: *"How does the character's body language support their angry state?"*), we modify only the corresponding input(s) of modality $m$ (i.e. visual) in the rejected pair, thereby enforcing that the model's response remains grounded in that modality. Thus, our prompt-based modality preference objective becomes,

$$\mathcal{L}_{\text{DPO}}^{av-prompt} = -\mathbb{E}\big[\log \sigma(u(a_w, v_w, \boxed{a_l^{\text{PMP}}, v_l^{\text{PMP}}}, x^m, y_w))\big] \tag{5}$$

where $a_l^{\text{PMP}} = a_w$, iff $m = \mathcal{V}$ and $v_l^{\text{PMP}} = v_w$, iff $m = \mathcal{A}$. We perform multiple forms of negative sampling for constructing $(a_l, v_l)$ (see Section 5.2); however, because our task is emotion reasoning, the best results were achieved when we choose the rejected audiovisual input to be a sample with an emotion different from the chosen input $(a_w, v_w)$.

**Emotion-based Response Preference.** To mitigate spurious cue-emotion associations and hallucinations described in Section 1, for a given input $(a_w, v_w, x)$ we construct two rejected responses that are variations of the chosen response $y_w$, as illustrated in Fig. 4*(Bottom)*. Specifically, $y_l^{vr}$ includes an audio/visual cue that is relevant to the audiovisual input but does not explain the emotion, whereas $y_l^{er}$ introduces audio/visual cues related to the emotion but absent from the audiovisual input (hallucinated). Following Huang et al. (2025b), we assign weights to these rejected responses in the DPO loss in Eq. (3) as,

$$\mathcal{L}_{\text{DPO}}^{y} = -\mathbb{E}_{(a_w, v_w, x, y_w, y_l^{vr}, y_l^{er}) \sim \mathcal{D}_y^{\text{pref}}} \left[ \log \sigma \left[ \beta \left( \log \frac{\pi_\theta(y_w \mid a_w, v_w, x)}{\pi_{\text{ref}}(y_w \mid a_w, v_w, x)} - \sum_{i \in \{vr, er\}} \beta_i \log \frac{\pi_\theta(y_l^i \mid a_w, v_w, x)}{\pi_{\text{ref}}(y_l^i \mid a_w, v_w, x)} \right) \right] \right] \tag{6}$$

where $\beta_{er} + \beta_{vr} = 1$. This formulation establishes strong contrasts between chosen and rejected responses, encouraging the policy model to ground its outputs in correct and emotion-relevant audiovisual cues. Unlike Huang et al. (2025b), however, we do not include completely irrelevant responses as rejections in DPO based on empirical findings in Appendix E.6.

## 4.2 TEXT PRIOR DEBIASING (TPD)

Audiovisual LLMs have strong text priors that cause them to hallucinate and include cues in their response, which usually occur together (e.g., the presence of a crying person accompanied by the

sound of crying). To suppress such behaviour, we propose to penalize the reward $r(a, v, x, y)$ derived in Eq. (2) to generate the response using only text input as follows,

$$r(a, v, x, y) = \beta \log \frac{\pi_\theta(y \mid a, v, x)}{\pi_{\text{ref}}(y \mid a, v, x)} + \beta \log Z(a, v, x) - \gamma_{\text{TPD}} \log \pi_{\text{text}}(y \mid x) \quad (7)$$

where $\pi_{\text{text}}$ is a trained language model and $\gamma_{\text{TPD}}$ is a hyperparameter. In our experiments, we choose $\pi_{\text{text}}$ to be the language model backbone in $\pi_{\text{ref}}$. This penalty ensures that the responses that are explainable purely by text priors get discounted and responses supported by audio/video get relative credit. Plugging Eq. (7) in the Bradley Terry model results in the following objective,

$$\mathcal{L}_{\text{DPO-TPD}} = -\mathbb{E}_{(a,v,x,y_w,y_l) \sim \mathcal{D}^{\text{pref}}} \left[ \log \sigma \left( \beta \left( \log \frac{\pi_\theta(y_w \mid (a, v, x))}{\pi_{\text{ref}}(y_w \mid (a, v, x))} - \log \frac{\pi_\theta(y_l \mid (a, v, x))}{\pi_{\text{ref}}(y_l \mid (a, v, x))} \right) \right. \right.$$
$$\left. \left. - \gamma_{\text{TPD}} \left( \log \pi_{\text{text}}(y_w \mid x) - \log \pi_{\text{text}}(y_l \mid x) \right) \right) \right] \quad (8)$$

where $(a, v)$ denote $(a_w, v_w)$ for simplicity. During training, we stop gradients through $\pi_{\text{text}}$ as it is just used to identify the text priors that a language model has. To maintain the text-only capabilities of the language model backbone, we attach LoRA module (Hu et al., 2022) to it for training. To accommodate two rejected responses, we perform scaling similar to Eq. (6) on the rejected responses in the TPD term as described in Appendix C.1 (Eq. (8)) to get the final TPD objective $\mathcal{L}^y_{\text{DPO-TPD}}$. The final objective function of **AVEm-DPO** is as follows,

$$\mathcal{L}_{\text{AVEm-DPO}} = \mathcal{L}^y_{\text{DPO-TPD}} + \lambda_{av} \mathcal{L}^{av-prompt}_{\text{DPO}} \quad (9)$$

where $\lambda_{av}$ is a hyperparameter. Implementation details are present in Appendix C.3.

## 4.3 Preference Data

For AVEm-DPO training, we construct preference data using a pipeline similar to Fig. 3. This preference dataset is different from *EmoReAlM*, which we exclusively use for testing. We use MAFW (Liu et al., 2022) and a subset of MER2025 (Lian et al., 2025b) *Track-1 train set* as the source datasets to create preference samples. We prompt Gemini-2.5 Gemini-Team et al. (2025) to generate variations of the correct answers (chosen responses) to the questions where the audiovisual cue is altered to be either a spurious emotion-related video-relevant cue ($y_l^{vr}$) or a hallucinated cue related to the emotion present ($y_l^{er}$). Note that we do not perform any manual verification on the generated data, which still results in a performance gain demonstrating the efficiency of the proposed approach. Details in Appendix C.2.

## 5 Experiments

**Datasets & Metrics.** For EmoReAlM benchmark, we report the average accuracy per task for all the tasks. For tasks with *Yes/No* responses, we additionally report the precision, recall and F1 score following previous multimodal hallucination benchmarks (Sung-Bin et al., 2025; Li et al., 2023). Beyond *EmoReAlM*, we also evaluate on established emotion recognition datasets—DFEW (Jiang et al., 2020), RAVDESS (Livingstone & Russo, 2018), MER2023 (Lian et al., 2023a)—and the emotion reasoning dataset EMER (Lian et al., 2023b). None of these datasets is used in training to ensure zero-shot evaluation. Following prior work (Cheng et al., 2024; Han et al., 2025b), we report unweighted and weighted average recalls for DFEW and RAVDESS and weighted F1 for MER2023. For emotion reasoning, we adopt GPT-based evaluation (Cheng et al., 2024), comparing generated responses against ground truth. In addition to clue and label overlap, we assess two dimensions: (i) *spurious cue–emotion associations*, where irrelevant cues are linked to emotions, and (ii) *hallucinatory cues*, where non-existent audiovisual cues are fabricated. For all metrics, higher values indicate better performance. Further details are provided in Appendix D.1.

**Reference models.** We use two audiovisual MLLMs as reference – EmotionLLaMA (Cheng et al., 2024) and our own developed base model. Our model is similar to EmotionLLaMA in architecture with changes to the audio encoder (*whisper-large-v3*(Radford et al., 2023)) and video encoder (*LanguageBind* (Zhu et al., 2024)). For EmotionLLaMA, we remove the text (subtitle) input branch to be consistent with the other baselines and retrain the model on the original dataset without subtitles – denoted as **EmotionLLaMA$^\star$** (Cheng et al., 2024). More details in Appendix D.2.

Table 2: Zero-shot performance comparison of different methods on existing audiovisual emotion recognition benchmarks. Mod. are the modalities input to the model with the prompt. A: Audio, V:Video, T: Text Subtitles. ‡: evaluation without text subtitle input.

| Model | Mod. | DFEW | | RAVDESS | | MER2023 | EMER | | | |
| | | UAR | WAR | UAR | WAR | F1 | Clue | Label | Spurious | Halluc. |
|---|---|---|---|---|---|---|---|---|---|---|
| VideoLLaMA 2 | A,V | 43.65 | 48.66 | 41.81 | 31.62 | 50.79 | 3.82 | 3.80 | 4.25 | 4.23 |
| OLA | A,V | 38.17 | 41.73 | 27.45 | 22.11 | 55.82 | 3.80 | 3.33 | 3.93 | 4.22 |
| VITA-1.5 | A,V | 39.31 | 42.56 | 50.67 | 46.88 | 66.94 | 4.77 | 4.72 | 5.16 | 5.70 |
| Qwen-2.5 Omni | A,V | 46.94 | 54.34 | 32.88 | 28.05 | 79.72 | 5.85 | 6.78 | 6.39 | 6.21 |
| EmotionLLaMA | A,V,T | 45.59 | 59.37 | 28.20 | 29.24 | 90.36 | 6.03 | 6.99 | 5.89 | 5.26 |
| EmotionLLaMA‡ | A,V | 42.72 | 54.06 | 30.36 | 30.45 | 89.05 | 2.76 | 2.78 | 3.44 | 2.36 |
| MoSEAR | A,V,T | 44.48 | 56.60 | - | - | 90.27 | - | - | - | - |
| **Our base** | A,V | 56.78 | 60.14 | 53.59 | 53.01 | 89.19 | 5.63 | 6.45 | 5.41 | 5.19 |
| + Naive-DPO | | 55.67 | 59.90 | 53.63 | 52.94 | 88.59 | 5.81 | 6.30 | 5.96 | 5.48 |
| + Vista-DPO† | | 56.42 | 62.33 | 56.94 | 53.64 | 90.06 | 6.08 | 6.89 | 6.58 | 6.07 |
| **+ AVEm-DPO** | | **58.54** | **64.24** | **58.66** | **55.48** | **92.18** | **6.37** | **7.08** | **7.09** | **6.75** |
| **EmotionLLaMA★** | A,V | 54.89 | 58.26 | 52.59 | 48.12 | 90.01 | 5.78 | 6.21 | 5.36 | 5.23 |
| + Naive-DPO | | 54.97 | 58.12 | 52.69 | 49.01 | 89.35 | 5.89 | 6.35 | 5.89 | 5.62 |
| + Vista-DPO† | | 56.28 | 61.58 | 56.42 | 50.96 | 91.19 | 6.05 | 6.56 | 6.85 | 6.31 |
| **+ AVEm-DPO** | | 57.06 | 62.12 | 56.21 | 51.03 | 91.68 | 6.02 | 6.99 | 7.02 | 6.62 |

**Baseline Preference Optimization Approaches.** We compare with original **Naive-DPO** (Rafailov et al., 2023) using single rejected samples from our DPO data and modified Vista-DPO (Huang et al., 2025b) for audiovisual inputs – denoted as **Vista-DPO†** (Appendix D.3 for details).

## 5.1 EMOTION REASONING AND RECOGNITION RESULTS

**EmoReAlM Results.** Table 3 presents the performance of different approaches on the proposed *EmoReAlM* benchmark. AVEm-DPO achieves substantial gains over the reference models, demonstrating the effectiveness of multimodal preference optimization and text-prior debiasing. While the baselines perform strongly on basic reasoning tasks, Table 3 shows that they struggle on *Modality Agreement* and *Stress-Test* evaluations (Expanded table in Appendix E.1 and Table 13).

Table 3: Performance comparison of different methods on the proposed *EmoReAlM* Benchmark.

| Model | Reas. Basic | | Modality Agree. | Reas. - Stress | |
| | Audio Acc. | Visual Acc. | F1 | Audio F1 | Visual F1 |
|---|---|---|---|---|---|
| VideoLLaMA2 | 63.1 | 66.8 | 52.5 | 53.2 | 58.4 |
| OLA | 63.2 | 60.4 | 42.7 | 56.6 | 54.8 |
| VITA-1.5 | 63.1 | 84.3 | 30.2 | 52.8 | 56.3 |
| Qwen 2.5 Omni | 76.8 | 89.2 | 33.3 | 55.0 | 56.8 |
| **Our base** | 69.2 | 85.3 | 34.6 | 50.3 | 59.9 |
| + Naive-DPO | 71.3 | 85.9 | 41.6 | 54.8 | 65.9 |
| + Vista-DPO† | 72.4 | 87.8 | 52.1 | 73.6 | 86.7 |
| **+ AVEm-DPO** | **77.9** | **92.5** | **60.0** | **80.9** | **94.6** |
| **Emot.-LLaMA★** | 64.8 | 84.9 | 33.1 | 46.7 | 63.2 |
| + Naive-DPO | 67.2 | 85.7 | 42.8 | 52.6 | 67.6 |
| + Vista-DPO† | 69.0 | 86.9 | 40.9 | 68.6 | 87.3 |
| **+ AVEm-DPO** | 76.5 | 89.9 | 56.8 | 75.4 | 91.7 |

Notably, our preference optimization also surpasses Vista-DPO and Naive-DPO by significant margins. To further examine the bottlenecks in baseline models, Appendix E.1 reports results on samples probing spurious audiovisual–emotion correlations and hallucinated cues. For state-of-the-art systems such as Qwen 2.5 Omni (Xu et al., 2025b) and VITA-1.5 Fu et al. (2025), hallucination emerges as a more severe issue than spurious cue-emotion associations. Moreover, unlike findings from Sung-Bin et al. (2025), our results indicate that audio and visual hallucinations are equally prevalent in emotion reasoning tasks. Additionally, Table 13 shows the performance of video-only and audio-only baselines and reveals that multimodal inputs hurt reasoning capabilities.

**Emotion Recognition and Reasoning on Existing Benchmarks.** Table 2 (expanded in Appendix E.3) shows the performance on existing emotion benchmarks mentioned before. We can notice that our reference models outperform baselines, showing the efficacy of reference in understanding emotion. Moreover, preference tuning additionally boosts the performance, especially for emotion reasoning on

Table 4: User evaluation on EMER.

| Model | Emot.↑ | Assoc.↑ | Incons.↓ |
|---|---|---|---|
| VideoLLaMA 2 | 9.82% | 0.75% | 15.38% |
| OLA | 9.36% | 7.46% | 5.58% |
| VITA 1.5 | 11.60% | 17.25% | 6.04% |
| Qwen 2.5 Omni | 10.75% | 18.57% | 10.13% |
| EmotionLLaMA | 1.89% | 11.53% | 68.61% |
| Our + AVEm-DPO | 54.74% | 43.35% | 4.67% |

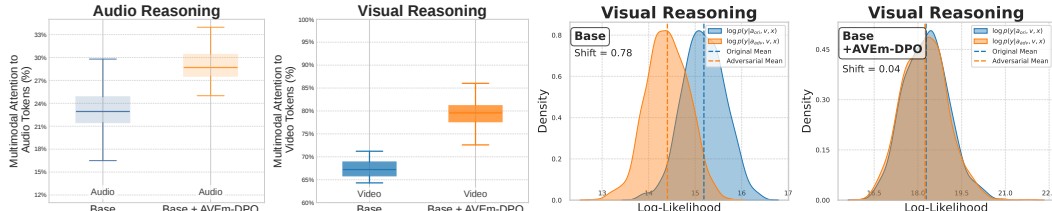

Figure 5: Effect of AVEm-DPO on – *(Left two plots)* the distribution of attention over video and audio tokens taken as a percentage over the total attention over all multimodal tokens for audio and visual reasoning tasks in *EmoReAlM*; *(Right two plots)* the log-likelihood distribution shift of the correct answer for visual reasoning tasks on corrupting the audio input $a_{ori}$ with adversary $a_{adv}$.

EMER, reducing spurious cue-emotion associ-
ations and hallucinations. It is important to note that previous emotion MLLM baselines (Cheng et al., 2024; Han et al., 2025b) use text subtitle as additional input. Qualitative comparison to baselines is present in Appendix F. While most baselines perform poorly on the out-of-domain RAVDESS dataset, our reference and preference-tuned models perform significantly better, showing their generalizability.

**User evaluation.** We perform a user evaluation with 40 participants on EMER generations from different models and report results in Table 4. Participants chose our model the most for emotion description and emotion-cue associations and the least for inconsistencies. (Details in Appendix E.4).

## 5.2 ANALYSIS

**Ablation Study.** Table 5 shows the performance of the preference-tuned model after removing the proposed components of AVEm-DPO. We perform this analysis on *EmoReAlM* and report the average metrics over audio and visual reasoning (Appendix D.5 for details). Removal of any of the key components results in a significant performance drop, especially for the reasoning tasks. Moreover, ablating TPD results in a huge performance drop on the hallucination stress test samples, underlining its efficacy in eliminating cue hallucinations in audio-visual emotion reasoning.

Table 5: Ablation study over different components of the proposed AVEm-DPO approach. PMP: Prompt-based Modality Preference, ERP: Emotion-based Response Preference, TPD: Text Prior Debiasing.

| Method | Basic. | Agree. | Stress | Spur. | Hall. |
|---|---|---|---|---|---|
| Our base | 77.3 | 34.6 | 55.1 | 47.3 | 39.2 |
| + AVEm-DPO | 85.2 | 60.1 | 87.8 | 92.7 | 97.6 |
| w/o PMP | 81.0 | 54.9 | 79.6 | 86.2 | 88.1 |
| w/o ERP | 81.8 | 56.2 | 79.4 | 84.9 | 88.4 |
| w/o TPD | 83.8 | 58.9 | 78.8 | 87.1 | 77.8 |
| + Contr. Dec. | 79.1 | 51.3 | 61.7 | 50.9 | 54.8 |

**Comparison with training-free contrastive decoding.** Similar to VCD Leng et al. (2024), we perform contrastive decoding using diffused audiovisual inputs and report results in Table 5 (*last row*), showcasing it is significantly worse than AVEm-DPO.

**Design Choices and Sensitivity to Hyperparams.** Appendix E.5 shows that prompt-based modality preference using a different emotion audiovisual (AV) input as $(a_l, v_l)$ works better compared to using random videos or diffused versions of the inputs. Appendix E.6 shows that using emotion-relevant and video-relevant rejected responses $(y_l^{er}, y_l^{vr})$ works better compared to only using one or using a completely irrelevant response. Appendix E.7 detail the sensitivity of AVEm-DPO to various hyperparameters, highlighting the role of various components in eliminating spurious cue-emotion associations and hallucinations.

**Attention redistribution after AVEm-DPO.** To analyze the effect of preference optimization on model attention, we plot the distribution of aggregate multimodal input attention over audio and visual tokens averaged over all attention heads for audio and visual reasoning tasks in *EmoReAlM* in Fig. 5 (*left two plots*). We can observe that the attention over relevant modality increases after AVEm-DPO, ensuring consistent model responses grounded on the relevant modality. More attention redistribution experiments are present in Appendix E.8.

**Robustness to adversarial inputs.** As shown in Fig. 12 (Appendix E.9), the model response on a prompt relevant to one modality should not change on changing the input of the irrelevant modality.

To test this robustness on visual reasoning tasks, we plot the distribution of log-likelihoods of correct responses for our base and AVEm-DPO models and show the distribution shift using Kernel Density Estimation (KDE) on changing the audio input in Fig. 5(*right two plots*). AVEm-DPO trained model results in negligible shifts, showing its robustness. Detailed analysis in Appendix E.9.

## 5.3 VALIDITY OF GENERATED PREFERENCE DATA

As mentioned in Section 4.3, our preference dataset is automatically generated using Gemini 2.5 (Gemini-Team et al., 2025). Performing human verification on the entire training data is too costly. Therefore, to show the validity of the generated preference tuning data, we perform human verification on a subset of 1000 random samples from the generated data with the help of 90 participants recruited through

Table 6: Human verification statistics on generated preference data.

| Response type | # Total verified | # Majority correct | # One or more correct |
|---|---|---|---|
| Chosen ($y_w$) | 1000 | 912 | 967 |
| Rejected - Video Relevant ($y_l^{vr}$) | 1000 | 895 | 923 |
| Rejected - Emotion Relevant ($y_l^{er}$) | 1000 | 856 | 912 |

Prolific (Prolific). Each generated sample is verified by three or more annotators. As shown in Table 6, for the different categories of preference responses mentioned in Section 4.1 – chosen ($y_w$), video-relevant rejected ($y_l^{vr}$), and emotion-relevant rejected ($y_l^{er}$) – we report the number of samples in which the majority of annotators found the generated responses correct. These results validate our automatically generated preference data.

## 6 LIMITATIONS AND FUTURE WORK

The proposed EmoReAlM benchmark is derived from the DFEW (Jiang et al., 2020) dataset, leveraging its emotion labels, and hence, it may inherit its cultural biases. Additionally, since our benchmark and training data are derived from existing emotion recognition datasets with short videos ($\sim$ 2-10 seconds), long video emotion understanding and reasoning remain an open topic that can be addressed in future work.

Although the proposed AVEm-DPO significantly improves the reference model's performance, a few limitations remain. Similar to other baselines, our model trained with AVEm-DPO performs poorly on the recognition for *disgust* (an ambiguous emotion (Hendel et al., 2023)) as shown in Appendix E.3 and Table 15. We attribute this to the limited amount of training samples available for this emotion class. Moreover, a closer look at the performance on the subtasks of the *Emotion Reasoning - Stress Test* task of EmoReAlM (Appendix E.2 and Table 14) reveals that there is still room for improvement to mitigate spurious audio cue-emotion associations.

## 7 CONCLUSION

This work addresses the bottlenecks of emotion reasoning in MLLMs, with two major contributions – *EmoReAlM* Benchmark for evaluating emotion reasoning over a complex and diverse set of tasks and *AVEm-DPO* preference optimization technique to mitigate bottlenecks of MLLMs such as spurious audiovisual cue-emotion associations and audiovisual cue hallucinations. The proposed method outperforms open-source baselines on the proposed and existing emotion understanding benchmarks under a zero-shot setting. Moreover, a detailed ablation study with analysis of attention redistribution and log-likelihood shift upon preference tuning supports the efficacy of the proposed prompt-based modality preference and text-prior debiasing approaches.

### ETHICS STATEMENT

This work builds upon publicly available audiovisual datasets for research purposes, specifically DFEW for benchmark creation (Section 3.2) and MAFW/MER2025 for preference optimization (Section 4.3). We did not collect new audiovisual data, ensuring no additional privacy risks. All data usage complies with the licensing terms of the original datasets. To mitigate potential harms, the released *EmoReAlM* benchmark will only contain automatically generated and human-verified question–answer pairs; users must independently obtain the underlying videos from the original sources under appropriate licenses. For human verification (Section 3.3) and user studies (Table 4),

participants were recruited via Prolific and compensated at fair rates commensurate with task requirements and participant location, aligning with ethical standards for crowd work. We ensured informed consent, anonymity and the right to withdraw at any point. The proposed methods aim to improve reliability in emotion reasoning by reducing hallucinations and spurious cue associations in multimodal large language models. However, emotion recognition and inference from audiovisual data can carry risks of misinterpretation, bias reinforcement, or misuse in surveillance and high-stakes applications. Moreover, users of the proposed method are advised to read the limitations of the proposed approach mentioned in Section 6 to avoid potential safety concerns. We emphasize that our benchmark and models are intended strictly for academic research, with the goal of advancing robust, interpretable and socially responsible AI. We caution against deployment in sensitive real-world contexts (e.g., healthcare, hiring, law enforcement) without careful domain-specific validation and safeguards.

## REPRODUCIBILITY STATEMENT

To ensure reproducibility and transparency, we provide additional details about data creation and experiments in the Appendix. All the prompts used for data creation are present in Appendix B.1. Implementation details for the proposed method, along with hyperparameter settings, are provided in Appendices C.3 and D.2, while the details about the baseline approaches are present in Appendices D.3 and D.4. Details about human verification of the benchmark and user evaluation are present in Appendices B.2 and E.4. Evaluation metrics are detailed in Appendix D.1. We also provide the detailed setup for our ablations in Appendix D.5. Our benchmark, code and model weights will be made publicly available upon acceptance to ensure reproducibility and ease of use for the proposed work. Code, models and benchmark at our project page - avere-iclr.github.io.

## ACKNOWLEDGEMENTS

Research was sponsored by the Army Research Office and was accomplished under Cooperative Agreement Number W911NF-25-2-0040. Work was also in part supported by the National Science Foundation under Grant IIS-2211550 and the National Institute of Mental Health of the National Institutes of Health under Award Number R61MH135407. The views and conclusions contained in this document are those of the authors and should not be interpreted as representing the official policies, either expressed or implied, of the Army Research Office, NSF, NIH, or the U.S. Government. The U.S. Government is authorized to reproduce and distribute reprints for Government purposes notwithstanding any copyright notation herein.

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

# APPENDIX

## TABLE OF CONTENTS

## A   LLM USAGE

We used GPT-5 to polish the text we added to the paper for grammar and consistency checks. We verify the grammar changes suggested by GPT to ensure its validity. No significant part of the text in the paper is written by any LLM. Apart from polishing the paper, we use LLMs for data annotation and automatic evaluation as mentioned in Sections 3.2 and 4.3 and Appendices C.2, D.1 and D.2.

## B   BENCHMARK DETAILS

### B.1   PROMPTS USED IN BENCHMARK CREATION

In this section, we detail the prompts that are used in various parts of the benchmark creation pipeline mentioned in Section 3 and Fig. 3. Note that the text prompts themselves are present at the end of the document in Appendix G.

Table 7: Statistics of human verification on *EmoReAlM* Benchmark.

| Task | | # Ques. verified | # Ques. at least one correct | # Ques. majority correct | # Ques discre. | # Ques. Final |
|---|---|---|---|---|---|---|
| Reasoning - Basic | Audio | 1200 | 1168 | 968 | 8 | 972 |
| | Visual | 1200 | 1137 | 1014 | 10 | 1024 |
| Modality Agreement | | 1000 | 489 | 458 | 0 | 456 |
| Reasoning - Stress Test | Audio | 1000 | 956 | 806 | 14 | 820 |
| | Visual | 1000 | 845 | 719 | 9 | 728 |
| **Total** | | **5400** | **4595** | **3959** | **41** | **4000** |

**Audio and Video Captioning.** Figs. 19 and 20 contains the prompts used to caption the audio and visual content separately for a given video as described in Section 3.2 and Fig. 3. For visual captioning, we sample eight uniform frames from the video and pass those to GPT-4o. For audio captioning, we only pass the audio as a WAV file to GPT-4o-audio.

**Emotion prediction from audio and video captions separately.** Figs. 21 and 22 contain prompts used to predict the emotion (out of the seven basic categories) just using the audio and video captions separately. If the ground truth emotion label cannot be predicted by both the audio and video captions, then we do not proceed with such a video for the subsequent data pipeline.

**EmoReAlM QA Generation.** Figs. 23 and 24 contains the prompts to generate questions related to *Emotion Reasoning - Basic* as described in Section 3.1 for audio and visual reasoning respectively. We use the ground truth emotion label already present in the source emotion recognition dataset, as well as the audio/video captions, to generate the question answers. Note that audio and visual reasoning samples are only generated for those samples in which emotion was predicted correctly from the audio and visual captions, respectively (using prompts in Figs. 21 and 22).

We use prompt in Fig. 25 to generate questions related to *Modality Agreement* (Section 3.1) by passing the audio captions, video captions and the ground truth emotion label present in the source dataset. We also verify the answers to the generated questions using the ground truth emotion label present for the video and the emotions predicted using only audio and video captions. If both the audio and the video caption predict the ground truth emotion label from the captions (using prompts in Figs. 21 and 22), then the correct answer should be *"Yes"*, else it should be *"No"*.

For the *Emotion Reasoning - Stress Test* (Section 3.1), we generate questions using prompts present in Figs. 26 to 31. We use separate prompts for generating questions related for the different subtasks – *No hallucination* (Figs. 26 and 29), *Spurious Cue-Emotion Association* (Figs. 27 and 30) and *Emotion-relevant Hallucination* (Figs. 28 and 31). Note that the *No hallucination* prompts only apply to cases where the emotion prediction from the audio and/or visual captions using Figs. 21 and 22 is same as the ground truth emotion label.

**Text Only Guess - Post Processing.** We use the prompt in Fig. 32 to guess the correct answer for the generated question and answer choices using only the text (i.e., without audiovisual input). This is done as a post-processing step as described in Section 3.3 to ensure that the answer for the MCQA sample is not predictable using only the text inputs.

### B.2 HUMAN VERIFICATION

As mentioned in Section 3.3, we perform human verification for the generated QA samples to ensure high data quality by removing samples that contain some discrepancy. We conducted a survey using Qualtrics and recruited participants using the crowd-sourcing platform Prolific. In total, we conducted the survey on 471 participants and ensured that the participants were paid fairly for their time. To ensure participants are capable of answering the questions, we included a pre-survey to test their emotional intelligence. Moreover, we included attention checks using questions that are already verified by us to ensure the quality of the participant responses.

We conduct the survey as a MCQ task where the participants are shown the questions and the answer choices created in the benchmark and we ask them to choose the correct answer as shown in Fig. 6. Each participant was also shown a follow-up question after each question to flag the text present in

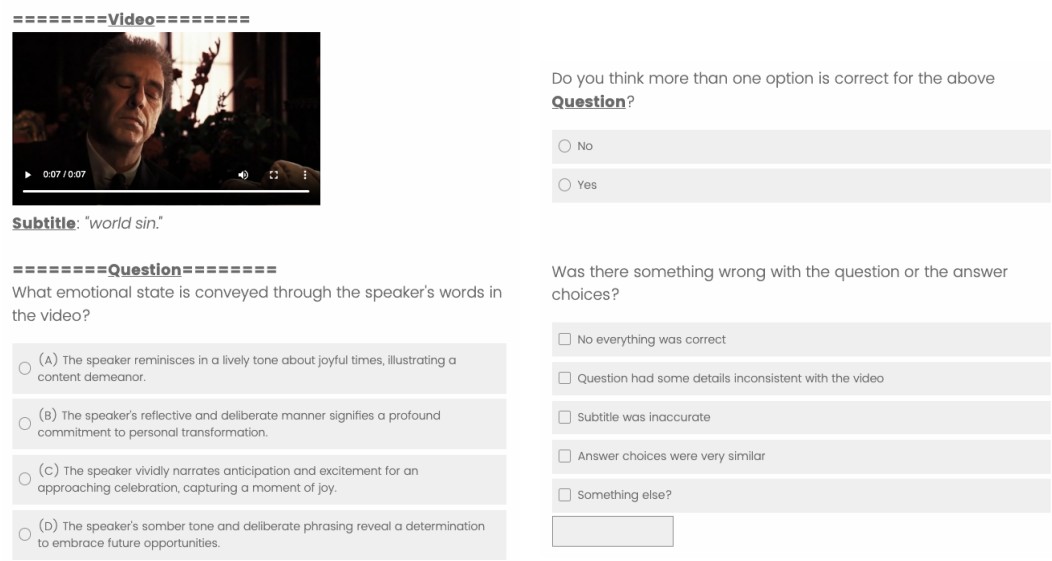

Figure 6: Human verification survey questions. *(Left)* An example question from the benchmark shown to the participant. *(Right)* Follow-up questions shown to the participant about each question.

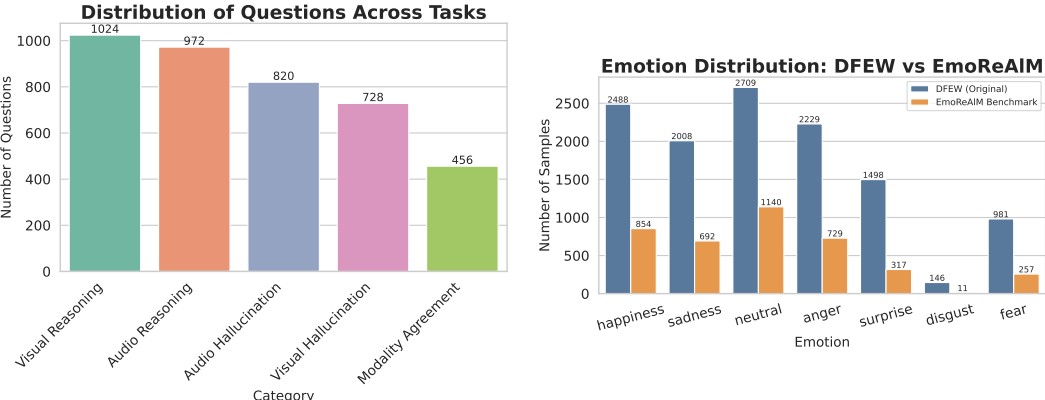

Figure 7: *(Left)* Distribution of QA samples across different tasks in EmoReAlM benchmark. *(Right)* Distribution of ground truth emotion labels for the videos present in EmoReAlM compared with the distribution in the source dataset DFEW (Jiang et al., 2020).

the question or answer choice or to report any other discrepancy. Since some videos in the DFEW (Jiang et al., 2020) dataset are not in English, the participants were also shown the English subtitle for the video that the MCQ is about.

Table 7 contains the statistics of human verification. Due to budget constraints, we ran the survey only on 5400 questions across different tasks. We only use the samples from the benchmark for which the majority of the participants selected the correct answer, automatically annotated in the benchmark. Additionally, we manually correct some samples that had discrepancies and add them to the final set of questions as well.

## B.3 BENCHMARK STATISTICS

Fig. 7 (*Right*) shows the distribution of ground truth emotion labels in the *EmoReAlM* benchmark compared to that present in the source dataset - DFEW (Jiang et al., 2020). We can see that the distribution of samples over different emotions is similar to DFEW. Fig. 9 shows the distribution of subtasks within the *Emotion Reasoning - Stress Test* task (Section 3.1) of *EmoReAlM* bench-

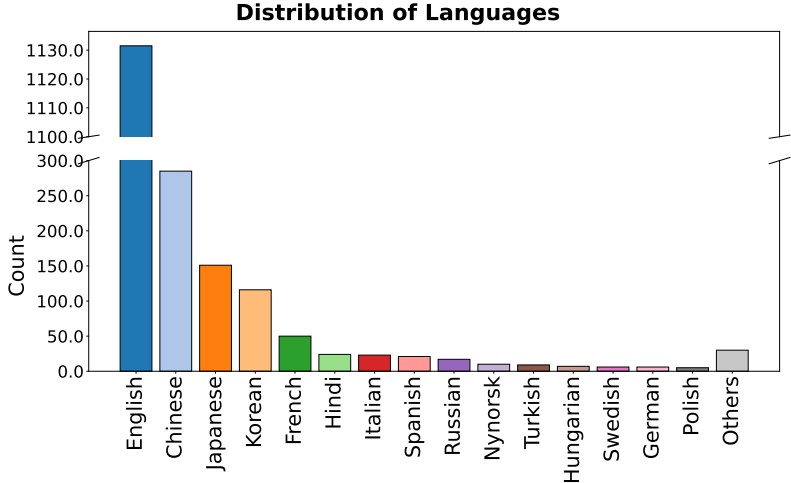

Figure 8: Distribution of different languages present in the audiovisual samples present in EmoRe-AlM benchmark.

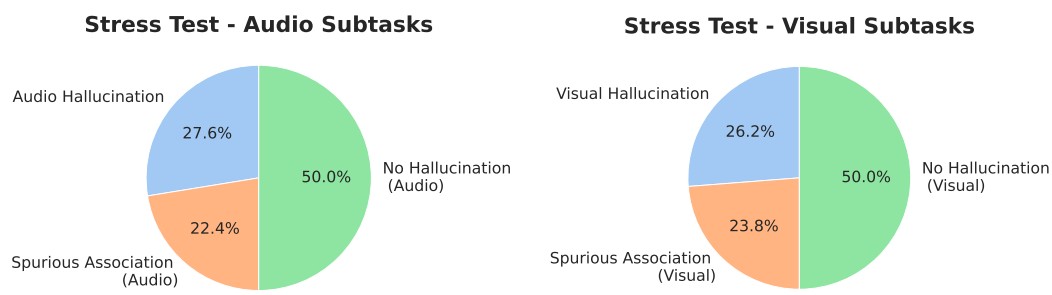

Figure 9: Distribution of subtasks in the *Emotion Reasoning - Stress Test* of *EmoReAlM* benchmark.

mark. Due to the way we formulate the questions for this subtask – *"Does the {audio/visual cue} suggest {emotion} of the character?"*, the samples belonging to *No hallucination* subtask have the answer *"Yes"*, and the samples in the *Spurious Association* and *Audio/Visual Hallucination* subtasks have answer *"No"*. Fig. 9 shows that the number of samples with *"Yes"*/*"No"* answers are equally distributed. Moreover, for all the samples with answers as *"No"*, the samples are almost equally distributed to test spurious cue-emotion associations and audiovisual cue hallucinations. Furthermore, to show the cultural and linguistic diversity in the benchmark, Fig. 8 shows the distribution of languages present in the samples of EmoReAlM benchmark. We obtain this by using automatic language detection using Whisper (Radford et al., 2023). We can observe that although the majority language is English, our benchmark contains samples from a wide range of languages.

Table 8: Effect of using different number of frames for visual captioning using GPT-4o.

| # frames | SBERT-sim | BERT Score | | |
| --- | --- | --- | --- | --- |
| | | Prec. | Rec. | F1 |
| 1 | 0.646 | 0.851 | 0.853 | 0.852 |
| 2 | 0.660 | 0.851 | 0.856 | 0.853 |
| 4 | 0.676 | 0.851 | 0.857 | 0.854 |
| 8 | 0.689 | 0.858 | 0.861 | 0.860 |
| 16 | 0.688 | 0.858 | 0.862 | 0.860 |

Table 9: Samples from the *EmoReAlM* Benchmark for the *Emotion Reasoning-Basic* Task.

| Task | Video | Question | Answer |
|------|-------|----------|--------|
| Reasoning Basic (Audio) | Subtitle: "I tried" | How does the speaker's choice of words in the video reflect their emotional state? (A) The speaker mentions struggling to move forward despite past setbacks, indicating a reflective state. (B) The speaker's tone reflects a somber atmosphere, accompanied by a soft, resigned voice. (C) The speaker's phrase portrays a deep sense of regret and resignation, reflecting a failed attempt. (D) The speaker uses soft background music to enhance the somber mood, suggesting unfulfilled efforts. | C |
| Reasoning Basic (Audio) | Subtitle: "You haven't spoken to me in 10 years" | In what way does the tone of the man's voice impact his emotional expression in the video? (A) The presence of soft whispers and gentle music in the background could imply an underlying tension and hidden emotion. (B) The man's tone is marked by a tightness and sharpness, resonating with his underlying frustration and simmering anger. (C) The phrase "I can't believe you've done this again" reflects an underlying resentment connected to a long-standing grievance. (D) The man's voice holds a lively and enthusiastic tone, mistakenly suggesting a sense of joy and contentment. | B |
| Reasoning Basic (Visual) | Subtitle: "Stanford University? What are you guys talking about?" | How does the woman's facial expression contribute to the overall feeling in the scene? (A) The woman displays a joyful expression with open arms, conveying her happiness and openness. (B) The woman's cheerful smile and lively eyes reveal her happiness and engagement. (C) The woman's yellow turtleneck adds a vibrant touch, symbolizing her happiness and contentment. (D) The woman's long dark hair frames her face, enhancing the appearance of happiness and delight. | B |
| Reasoning Basic (Visual) | Subtitle: "" | What does the individual's body language indicate about their emotional state in the video? (A) The individual's quivering movements and uncertain footing create a palpable sense of fear. (B) The person's tense facial expression with slightly open mouth and wide eyes enhances their fearful demeanor. (C) The person is leaning cautiously towards the door, their body tense, which highlights their fear or anxiety. (D) The individual's dark-colored shirt amplifies their sense of fear, overshadowing their surroundings. | C |

## B.4 FRAME SAMPLING RATE FOR AUTOMATIC VISUAL CAPTIONING

Since the visual cues used to express and infer emotions can be subtle, it is important to ensure that the visual captions obtained using GPT-4o in the first stage of data creation (Section 3.2 and Fig. 3) are of high quality. To identify the ideal number of frames to be sampled from the video for captioning, we ran a small experiment on the emotion captioning dataset EMER (Lian et al., 2023b). It is important to note that EMER (mean duration: 3.78s) contains videos of similar duration as DFEW (mean duration: 3.42s) which we use to construct EmoReAlM. We extract different number of frames per video and obtain the visual caption from GPT-4o using prompt in Fig. 20. Then, compute the similarity between the generated captions and the ground truth using BERTScore (Zhang et al.) and Sentence BERT (Reimers & Gurevych, 2019) similarity score. Table 8 shows that using 8 frames for visual captioning leads to good captioning results. Furthermore, using 16 frames is not significantly better than using 8 frames, but it increases the costs significantly. Hence we choose to use 8 frames uniformly sampled from the video to extract visual captions from GPT-4o automatically.

## B.5 BENCHMARK SAMPLES

We present samples belonging to different categories of the benchmark in Tables 9 to 11. Note that the subtitles shown in the tables are just for reference and we do not pass the subtitle as an input to the model during evaluation.

Table 10: Samples from the *EmoReAlM* Benchmark for the *Modality Agreement* Task.

| Task | Video | Question | Answer |
|---|---|---|---|
| Modality Agreement | Subtitle: "I was..." | Do the visual elements of the video align with the audio in conveying the feeling of happiness of the person in the video? 
 (A) Yes 
 (B) No | B |
| Modality Agreement | Subtitle: "That is exactly what I am" | Do the audio and video modalities align for the expression of anger of the person in the video? 
 (A) Yes 
 (B) No | A |

Table 11: Samples from the *EmoReAlM* Benchmark for the *Emotion Reasoning-Stress Test*.

| Task | Video | Question | Answer |
|---|---|---|---|
| Stress Test (Audio No Hallucination) | Subtitle: "(chuckles)" | Do the chuckling sounds in the audio enhance the feeling of joy conveyed for the person in the video? 
 (A) Yes 
 (B) No | A |
| Stress Test (Audio - Spurious Association) | Subtitle: "(sonar ping)" | Is the presence of a sonar ping sound effect crucial to the feeling of surprise conveyed by the person in the video? 
 (A) Yes 
 (B) No | B |
| Stress Test (Audio - Hallucination) | Subtitle: "It ain't Alan's fault..." | Does the sound of a slamming door contribute to the anger experienced by the person in the video? 
 (A) Yes 
 (B) No | B |
| Stress Test (Visual No Hallucination) | Subtitle: "" | Is the downward gaze of the older woman a significant factor in expressing the sadness of the older woman portrayed in the video? 
 (A) Yes 
 (B) No | A |
| Stress Test (Visual - Spurious Association) | Subtitle: "" | Is the presence of the vibrant checkered pattern on the walls a factor in conveying the neutral emotion of the person/character in the video? 
 (A) Yes 
 (B) No | B |
| Stress Test (Visual - Hallucination) | Subtitle: "" | Is the man displaying a clenched fist as a sign of his anger in this video? 
 (A) Yes 
 (B) No | B |

Table 12: Examples of the preference dataset used for AVEm-DPO.

| Video | Prompt ($x$) | Chosen Response ($y_w$) | Rejected Response (video-relevant - $y_l^{vr}$) | Rejected Response (emotion-relevant - $y_l^{er}$) |
|---|---|---|---|---|
| Subtitle: "You're a bully. But I can never fight back, because you are JJ!" | How do the facial expressions of the young person contribute to the emotional intensity during the exchange? | The young person's furrowed eyebrows and open mouth emphasize their intense emotional state and frustration. | The dark top worn by the young person underlines the seriousness of their mood. | The young person's hands clenching into fists and subtle scowling underline their frustration. |
| Subtitle: "I'm so tired." | How does the woman's message in the video reflect her emotional state? | She communicates a deep sense of exhaustion and emotional weariness through her words, saying 'I'm so tired,' which indicates her sadness. | The melancholic piano music in the background underscores the emotional heaviness she is experiencing. | Her loud expressive crying, typically associated with sadness, conveys the depth of her emotional state. |
| Subtitle: "(crying)" | Do the audio and video convey the same emotional state for the woman in the video? | Yes, both the audio and video convey a profound sense of sadness through the sounds of crying and the woman's distraught facial expression. | No, the tone of voice in the audio appears sad, but the stark background in the video suggests a more calm atmosphere. | No, the woman's facial expression indicates a sense of fear, while her words "I can not take it anymore" suggest sadness. |

# C METHODOLOGICAL DETAILS

## C.1 TEXT-PRIOR DEBIASING

Similar to Eq. (6), we scale the TPD term to accommodate multiple rejected responses as follows,

$$
\mathcal{L}_{\text{DPO-TPD}}^{y} = -\mathbb{E}_{(a,v,x,y_w,y_l)\sim\mathcal{D}^{\text{pref}}}\Bigg[\log\sigma\Bigg(\beta\Bigg(\log\frac{\pi_\theta(y_w\mid(a,v,x))}{\pi_{\text{ref}}(y_w\mid(a,v,x))} - \sum_{i\in\{vr,er\}}\beta_i\log\frac{\pi_\theta(y_l^i\mid(a,v,x))}{\pi_{\text{ref}}(y_l^l\mid(a,v,x))}\Bigg)
$$
$$
-\gamma_{\text{TPD}}\Big(\log\pi_{\text{text}}(y_w\mid x) - \sum_{i\in\{vr,er\}}\beta_i\log\pi_{\text{text}}(y_l^i\mid x)\Big)\Bigg)\Bigg]
$$

(10)

where $\beta_{vr} + \beta_{er} = 1$. Also, for succinctness, we denote $(a_w, v_w)$ with $(a, v)$ in the above equation.

## C.2 PREFERENCE DATA

As mentioned in Section 4.3, we use a pipeline similar to Fig. 3 to construct our preference data using MAFW (Liu et al., 2022) and MER2025 (Lian et al., 2025b) *Track 1 train set* as the source datasets. Note that we use Gemini 2.5 Flash (Gemini-Team et al., 2025) for all automatic annotations required to create the training dataset. Use of Gemini for training data creation reduces annotation budget and ensures that the training dataset is not biased to have similar language as the test dataset – *EmoReAlM*. Since the pipeline in Fig. 3 creates MCQA samples, we use another round of automatic annotations through Gemini-2.5 Flash over the generated MCQA samples to create the preference data. Specifically, we use prompts in Figs. 33 to 35 to generate rejected responses for the generated emotion reasoning QA samples. Since, we also want to improve the performance on emotion description tasks present in EMER (Lian et al., 2023b) we use prompts for audio (Fig. 33) and visual reasoning (Fig. 34) to modify emotion descriptions generated from Gemini 2.5 Flash (using prompt in Fig. 36), combining audio and visual captions of MAFW and MER2025 (obtained using prompts in Figs. 19 and 20). After Gemini annotation, we end up with a total of 41687 preference samples combining tasks, which we use for AVEm-DPO training. Table 12 contains samples from the constructed preference dataset using the described pipeline.

## C.3 IMPLEMENTATION DETAILS

We train the reference models using AVEm-DPO for one epoch, with a learning rate of $5e^{-7}$ and per GPU batch size of 2 on an NVIDIA DGX node with 8 NVIDIA H100 GPUs. We choose $\beta$ as 0.1 similar to (Huang et al., 2025b). Moreover, $\lambda_{av}$ is set to 1.0, $\beta_{er}$ and $\beta_{vr}$ are both set to 0.5, and $\gamma_{\text{TPD}}$ is set to 0.2 (refer to Appendix E.7 for details on choice). We attach LoRA module with

rank 8 and scale 4 to the LLM backbone for training. Gradient accumulation is used to accumulate gradients over 4 iterations.

# D EXPERIMENTAL DETAILS

## D.1 EVALUATION METRICS

**GPT Evaluation on EMER.** As mentioned in Section 5, we perform GPT-4o evaluation on the generated emotion descriptions in EMER (Lian et al., 2023b) dataset. We perform the evaluation over the following criterias – (i) *clue overlap* - similarity of the audiovisual cues present in the generation with the ground truth, (ii) *label overlap* - similarity of the emotion label described in the generation with the ground truth, (iii) *spurious cue-emotion associations* - how good are the audiovisual cues associated with emotions in the generation, and (iv) *hallucinatory cues* - presence of cues that are absent in the ground truth but present in the generations. The prompt used to evaluate the generations is present in Fig. 37.

**EmoReAlM Evaluation Metrics.** For all the tasks in *EmoReAlM*, we report the average accuracy over the task, computed as the number of correct responses out of the total number of samples in the task. Additionally, for tasks with *"Yes"/"No"* responses (*Modality Agreement* and *Emotion Reasoning - Stress Test*), we report the precision, recall and F1 score. Precision and recall are the ratios of correctly answered questions that have correct answers as *Yes* and *No*, respectively. F1 score is the harmonic mean of precision and recall.

## D.2 REFERENCE MODELS

We describe the reference models mentioned in Section 5 below.

**Our base.** We modify EmotionLLaMA (Cheng et al., 2024) to replace the visual encoder with LanguageBind Video Encoder (Zhu et al., 2024) and audio encoder with Whisper Large v3 (Radford et al., 2023). We pretrain the visual projector using the pretraining data of VideoLLaVA (Lin et al., 2024) and the audio projector is pretrained using LibriSpeech (Panayotov et al., 2015) and SpeechCraft (Jin et al., 2024) to enhance paralinguistic capabilities of the model. We finetune on the EmotionLLaMA dataset, however, we include additional instruction data by annotating MAFW (Liu et al., 2022) and MER2025 (Lian et al., 2025b) *Track 1 train set* through Gemini 2.5 Flash. Specifically, we use the prompts mentioned in Appendix B.1 to create a finetuning dataset with similar tasks as in the proposed EmoReAlM benchmark. We also use prompt in Fig. 36 to generate emotion descriptions from MAFW and MER2025.

**EmotionLLaMA⋆.** Since the pretrained EmotionLLaMA model is not trained on tasks similar to *EmoReAlM*, we finetune EmotionLLaMA on additional datasets created using MAFW and MER2025, similar to our base model described in the previous paragraph. Moreover, we do not provide subtitle text as input to the model during finetuning, in contrast to the original EmotionLLaMA, to eliminate external subtitle dependence.

## D.3 BASELINE PREFERENCE OPTIMIZATION APPROACHES

We describe the implementation of baseline DPO approaches mentioned in Section 5 below. We use the same training setup as mentioned in Appendix C.3 unless stated otherwise.

**Naive-DPO.** For Naive-DPO (Rafailov et al., 2023) we use the objective in Eq. (3). We use the preference samples from our preference data (Appendix C.2), and pick the rejected response randomly between $y_l^{vr}$ and $y_l^{er}$.

**Vista-DPO†.** We adapt Vista-DPO (Huang et al., 2025b) for audiovisual inputs using Eqs. (4) and (6). Also, we use our preference data (Appendix C.2) to optimize Eq. (4) and drop their temporal (clip-based) and object-based preferences. Instead of prompt-based modality preference, we use $(a_l, v_l)$ to be an audiovisual input that has a different emotion than that of $(a_w, v_w)$, always irrespective of the input prompt.

## D.4 BASELINE IMPLEMENTATIONS

**Audiovisual baselines.** We use the official code for Qwen 2.5 Omni - 7B (Xu et al., 2025a) and run inference using flash attention 2. We use their default system prompt during inference.

For Video-LLaMA (Zhang et al., 2023), we use the official video-language checkpoint *finetune-vicunna7b-v2* and audio-language checkpoint *finetune-vicuna7b-audiobranch*. We also use the default conversation template for inference.

For PandaGPT (Su et al., 2023), we use their official pretrained checkpoint *pandagpt-7b* with 1,024 *max_len*, built upon ImageBind (Girdhar et al., 2023). The system prompt remains unchanged during inference.

For OneLLM (Han et al., 2025a), we use the released pretrained checkpoint *OneLLM-7B*; for inference, we manually prepend the multimodal representations before the textual prompt.

We use VITA-1.5 (Fu et al., 2025) with its official code and checkpoint, including the *InternViT-300M* vision tower and the pretrained audio encoder. We use the default conversation template for inference.

**Audio-only baselines.** We use the official *Qwen2-Audio-7B-Instruct* (Chu et al., 2024) checkpoint and its default conversation template with the original system prompt.

For Kimi-Audio (Ding et al., 2025), we use the released *Kimi-Audio-7B-Instruct* checkpoint with the default system message.

For Audio Flamingo 3 (Goel et al., 2025), we use the official repository, pretrained checkpoint, and the default empty conversation template.

**Video-only baselines.** We use the official code for InternVL3.5 (Wang et al., 2025). Unlike others, this is an 8B model.

For Qwen2.5-VL (Bai et al., 2025), we use the released *Qwen2.5-VL-7B-Instruct* checkpoint with the default system prompt.

For *VideoLLaMA3-7B* (Zhang et al., 2025a), we used the default system message and run inference with flash attention 2.

## D.5 EXPERIMENTAL SETUP FOR ABLATION STUDY

We describe the setup for the ablations mentioned in Section 5.2 in detail below.

For Tables 5 and 17 and Fig. 11, the metric reported for *Emotion Reasoning – Basic* (denoted as **Basic**) is the unweighted average of the visual and audio reasoning accuracy on the *Emotion Reasoning – Basic* task. For *Emotion Reasoning – Stress Test* (denoted as **Stress**), the reported metric is the unweighted average of the F1 scores for visual and audio reasoning samples within the *Emotion Reasoning – Stress Test* task. For *Modality Agreement* (denoted as **Agree**), we report the F1 score over samples from the *Modality Agreement* task. Additionally, for the subtasks *Spurious Cue–Emotion Association* (denoted as **Spur.**) and *Emotion-Relevant Cue Hallucination* (denoted as **Hall.**), we use the unweighted average accuracy across visual and audio reasoning samples for each respective subtask.

**Ablation Study.** For Table 5, the model without prompt-based modality preference (w/o PMP) is trained only using $\mathcal{L}^y_{\text{DPO-TPD}}$ (Eq. (10)). The model without emotion-based response preference (w/o ERP) is trained using the the following loss,

$$\mathcal{L}_{\text{w/o ERP}} = \mathcal{L}_{\text{DPO-TPD}} + \mathcal{L}_{\text{DPO}}^{av-prompt} \tag{11}$$

refer Eqs. (5) and (8) for the involved terms. Finally, the model without text prior debiasing (w/o TPD) is trained on the following objective,

$$\mathcal{L}_{\text{w/o TPD}} = \mathcal{L}_{\text{DPO}}^y + \mathcal{L}_{\text{DPO}}^{av-prompt} \tag{12}$$

refer Eqs. (5) and (6) for the involved terms.

Table 13: Performance comparison of different methods on the proposed EmoReAlM Benchmark. **Bold** are best results and underline are second-best results over open-source models.

| Model | Reas. Basic | | Modality Agreement | | | | Reasoning - Stress Test | | | | | | | | Avg. Acc. |
|---|---|---|---|---|---|---|---|---|---|---|---|---|---|---|---|
| | Audio Acc. | Visual Acc. | Acc. | Pre. | Rec. | F1 | Audio | | | | Visual | | | | |
| | | | | | | | Acc. | Pre. | Rec. | F1 | Acc. | Pre. | Rec. | F1 | |
| *Closed-source models* | | | | | | | | | | | | | | | |
| Gemini 2.5 Flash | 78.0 | 88.9 | 57.0 | 75.9 | 39.0 | 51.5 | 63.5 | 74.0 | 51.0 | 60.4 | 73.2 | 75.3 | 70.9 | 73.0 | 72.1 |
| Gemini 2.5 Pro | 72.7 | 87.0 | 54.7 | 76.0 | 33.3 | 46.3 | 63.8 | 74.0 | 53.3 | 62.0 | 73.1 | 84.0 | 59.8 | 69.8 | 70.3 |
| *Open-source video-only models* | | | | | | | | | | | | | | | |
| VideoLLaMA 3 | - | 86.2 | - | - | - | - | - | - | - | - | 64.9 | 97.9 | 33.0 | 49.4 | - |
| Qwen 2.5 VL | - | 88.1 | - | - | - | - | - | - | - | - | 75.2 | **98.6** | 52.6 | 68.5 | - |
| InternVL 3.5 | - | **92.8** | - | - | - | - | - | - | - | - | 68.3 | 91.6 | 45.8 | 61.1 | - |
| *Open-source audio-only models* | | | | | | | | | | | | | | | |
| Qwen 2 Audio | 56.6 | - | - | - | - | - | 55.1 | 84.2 | 28.3 | 42.3 | - | - | - | - | - |
| Kimi-Audio | 69.8 | - | - | - | - | - | 54.0 | 95.8 | 15.5 | 26.6 | - | - | - | - | - |
| Audio Flamingo 3 | 76.8 | - | - | - | - | - | 52.6 | **96.7** | 11.9 | 21.2 | - | - | - | - | - |
| *Open-source audiovisual ("omni") models* | | | | | | | | | | | | | | | |
| VideoLLaMA | 21.7 | 22.2 | 34.1 | 37.4 | 30.9 | 33.9 | 46.1 | 41.3 | 50.6 | 45.5 | 48.8 | 48.4 | 49.2 | 48.8 | 37.1 |
| PandaGPT | 37.4 | 35.7 | 53.7 | 50.3 | **56.9** | 53.4 | 45.8 | 62.9 | 30.1 | 40.7 | 47.1 | 59.9 | 34.7 | 43.9 | 44.0 |
| OneLLM | 42.0 | 55.6 | 54.8 | 64.3 | 45.9 | 53.5 | 56.8 | 87.1 | 28.9 | 43.4 | 62.0 | 97.6 | 27.6 | 43.1 | 54.2 |
| VideoLLaMA2 | 63.1 | 66.8 | 52.6 | 52.0 | 53.0 | 52.5 | 53.7 | 60.6 | 47.3 | 53.2 | 59.4 | 67.9 | 51.2 | 58.4 | 59.1 |
| OLA | 63.2 | 60.4 | 51.7 | 78.9 | 29.8 | 42.7 | 63.5 | 86.8 | 41.9 | 56.6 | 62.3 | 85.0 | 40.4 | 54.8 | 60.2 |
| VITA-1.5 | 63.1 | 84.3 | 51.7 | 87.1 | 18.2 | 30.2 | 63.0 | 91.0 | 37.2 | 52.8 | 66.1 | 92.7 | 40.4 | 56.3 | 65.6 |
| Qwen 2.5 Omni | 76.8 | 89.2 | 52.2 | 86.1 | 37.2 | 33.3 | 63.0 | 91.4 | 39.6 | 55.0 | 67.8 | 96.4 | 40.3 | 56.8 | 70.0 |
| **Our base** | 69.2 | 85.3 | 51.4 | 86.3 | 21.6 | 34.6 | 53.1 | 65.4 | 40.8 | 50.3 | 66.4 | 87.2 | 45.6 | 59.9 | 65.1 |
| + Naive-DPO | 71.3 | 85.9 | 57.3 | 87.2 | 27.3 | 41.6 | 55.6 | 62.3 | 48.9 | 54.8 | 70.6 | 88.8 | 52.4 | 65.9 | 68.1 |
| + Vista-DPO[†] | 72.4 | 87.8 | 63.1 | 89.4 | 36.8 | 52.1 | 74.1 | 67.8 | 80.4 | 73.6 | 87.0 | 92.1 | 81.9 | 86.7 | 76.9 |
| **+ AVEm-DPO** | 77.9 | 92.5 | **68.9** | 93.4 | 44.3 | **60.0** | **82.6** | 70.7 | 94.6 | **80.9** | 94.6 | 93.1 | **96.1** | **94.6** | **83.3** |
| Δ% (relative) | 12.6 | 8.4 | 34.1 | 8.2 | 105. | 73.4 | 55.6 | 8.1 | 131. | 60.8 | 42.5 | 6.8 | 110. | 57.9 | 28.0 |
| Emot.-LLaMA[⋆] | 64.8 | 84.9 | 51.2 | 82.9 | 20.7 | 33.1 | 48.9 | 59.2 | 38.5 | 46.7 | 69.1 | 89.3 | 48.9 | 63.2 | 63.6 |
| + Naive-DPO | 67.2 | 85.7 | 56.1 | 83.4 | 28.8 | 42.8 | 53.5 | 60.1 | 46.8 | 52.6 | 71.9 | 89.5 | 54.3 | 67.6 | 66.9 |
| + Vista-DPO[†] | 69.0 | 86.9 | 58.2 | 85.9 | 30.4 | 40.9 | 69.2 | 63.1 | 75.2 | 68.6 | 87.6 | 92.5 | 82.6 | 87.3 | 74.2 |
| **+ AVEm-DPO** | 76.5 | 89.1 | 65.6 | 89.5 | 41.6 | 56.8 | 77.3 | 65.2 | 89.4 | 75.4 | 91.8 | 92.6 | 90.9 | 91.7 | 80.1 |
| Δ% (relative) | 18.1 | 4.9 | 28.1 | 8.0 | 101. | 71.6 | 58.1 | 10.1 | 132. | 61.5 | 32.9 | 3.7 | 85.9 | 45.1 | 25.5 |

# E DETAILED RESULTS

## E.1 EMOReALM RESULTS - EXPANDED

Table 13 shows the the expanded version of Table 3 with accuracy, precision and recall metrics for *Modality Agreement* and *Emotion Reasoning - Stress Test* categories. We also report the unweighted average accuracy over all five tasks in the benchmark in the last column. The relative percent improvement of the AVEm-DPO trained model over the reference models is present as the Δ% row. Moreover, we also report the performance of video-only and audio-only baselines in Table 13. We can see that for visual reasoning tasks (*Basic* and *Stress Test*), video-only baselines perform slightly better than the audiovisual ("*omni*") baselines, aligning with the findings of Sung-Bin et al. (2025). However, for audio reasoning tasks, audiovisual baselines outperform audio-only baselines, which have very poor recall on the *Emotion reasoning - Stress Test*. This can be attributed to the limited amount of audio-emotion datasets that the baselines (Chu et al., 2024; Ding et al., 2025; Goel et al., 2025) are trained on resulting in poor emotion reasoning.

## E.2 EMOReALM RESULTS ON DIFFERENT STRESS TEST SUBTASKS

Table 14 shows the performance of different baselines as well as AVEm-DPO on different subtasks of *Emotion Reasoning - Stress Test*, which have answer as *"No"* – *Spurious Cue-Emotion Association* and *Emotion-relevant Cue Hallucination* (refer to Section 3.1 and Appendix B for definitions). We can observe that within audio and visual reasoning, hallucination seems to be a bigger bottleneck than spurious cue-emotion associations. Moreover, similar to Table 13, we can observe that the audio-only models perform worse compared to audiovisual models, whereas the video-only model performance is better compared to audiovisual models. AVEm-DPO improves the model performance over all the subtasks significantly compared to the reference model.

Table 14: Performance of different baselines on different Reasoning Stress-Test sub-tasks in Emo-ReAlM Benchmark. This experiment is done only using samples from the Stress-Test category of the benchmark which have correct answer as "No". **Bold** are best results and underline are second-best results over open-source models.

| Model | Audio | | Visual | |
|---|---|---|---|---|
| | Spur. | Hall. | Spur. | Hall. |
| *Open-source video-only models* | | | | |
| VideoLLaMA 3 | - | - | 37.4 | 29.1 |
| Qwen 2.5 VL | - | - | 64.7 | 41.8 |
| InternVL 3.5 | - | - | 50.4 | 41.8 |
| *Open-source audio-only models* | | | | |
| Qwen 2 Audio | 41.8 | 16.9 | - | - |
| Kimi Audio | 26.8 | 6.0 | - | - |
| Audio Flamingo 3 | 15.7 | 8.7 | - | - |
| *Open-source audiovisual ("omni") models* | | | | |
| VideoLLaMA | 27.5 | 35.0 | 33.1 | 37.4 |
| PandaGPT | 43.1 | 19.1 | 47.5 | 23.4 |
| OneLLM | 47.7 | 13.1 | 36.7 | 19.6 |
| VideoLLaMA2 | 61.4 | 35.5 | 57.6 | 45.6 |
| OLA | 52.9 | 32.8 | 56.8 | 25.9 |
| VITA-1.5 | 46.4 | 29.5 | 46.0 | 35.4 |
| Qwen 2.5 Omni | 53.4 | 28.1 | 51.9 | 30.1 |
| **Our base** | 45.2 | 36.4 | 49.3 | 41.9 |
| + Naive-DPO | 49.9 | 47.9 | 56.8 | 48.0 |
| + Vista-DPO[†] | 85.7 | 75.1 | 87.1 | 76.7 |
| + AVEm-DPO | **88.6** | **99.5** | **96.5** | **95.8** |

Table 15: Class-wise recall for different emotion classes in DFEW dataset. **Bold** are best results and underline are second-best results over open-source models.

| Model | Mod. | Hap. | Sad. | Neu. | Ang. | Sur. | Dis. | Fea. | UAR | WAR |
|---|---|---|---|---|---|---|---|---|---|---|
| *Open-source video-only models* | | | | | | | | | | |
| VideoLLaMA 3 | V | 77.92 | 41.38 | 40.88 | 42.53 | 26.44 | **34.26** | 72.30 | 47.96 | 49.47 |
| Qwen 2.5 VL | V | 64.21 | 52.37 | **69.49** | 39.09 | 11.38 | 7.20 | 75.03 | 45.54 | 52.32 |
| InternVL 3.5 | V | 79.49 | 77.20 | 45.42 | 21.38 | 53.02 | 12.61 | 62.10 | 50.18 | 55.46 |
| *Open-source audio-only models* | | | | | | | | | | |
| Qwen 2 Audio | A | 64.55 | 25.08 | 2.28 | 0.00 | 0.06 | 2.07 | 53.55 | 21.08 | 22.24 |
| Kimi Audio | A | 50.34 | 42.97 | 37.50 | 71.24 | 12.66 | 10.34 | 29.93 | 36.43 | 43.30 |
| Audio Flamingo 3 | A | 2.98 | 19.96 | 12.92 | 83.01 | 6.12 | 15.86 | 41.46 | 26.05 | 26.39 |
| *Open-source audiovisual ("omni") models* | | | | | | | | | | |
| PandaGPT | A,V | 60.50 | 9.95 | 0.0 | 58.61 | 0.00 | 0.00 | 0.00 | 18.44 | 24.20 |
| VideoLLaMA | A,V | 85.04 | 8.41 | 4.17 | 20.84 | 3.95 | 0.00 | 1.14 | 17.65 | 24.09 |
| OneLLM | A,V | 47.91 | 54.33 | 3.23 | 52.35 | 26.08 | 1.80 | 70.21 | 36.74 | 37.60 |
| VideoLLaMA2 | A,V | **87.50** | 57.93 | 7.94 | 58.56 | 42.08 | 15.00 | 36.54 | 43.65 | 48.66 |
| OLA | A,V | 52.00 | **82.20** | 15.65 | 48.95 | 9.65 | 10.00 | 48.72 | 38.17 | 41.73 |
| VITA-1.5 | A,V | 61.46 | 79.96 | 23.54 | 23.19 | 8.05 | 0.90 | **78.07** | 39.31 | 42.56 |
| Qwen 2.5 Omni | A,V | 45.45 | 73.84 | 61.11 | 70.64 | 4.40 | 0.00 | 73.15 | 46.94 | 54.33 |
| EmotionLLaMA | A,V,T | 71.98 | 76.25 | 61.99 | 71.95 | 33.67 | 0.00 | 3.31 | 45.59 | 59.37 |
| MoSEAR | A,V,T | 79.35 | 75.20 | 40.45 | 69.66 | 42.86 | 0.00 | 3.87 | 44.48 | 56.60 |
| Our base | A,V | 70.75 | 72.07 | 29.64 | **77.04** | 61.54 | 27.59 | 58.87 | 56.78 | 60.14 |
| +AVEm-DPO | A,V | 75.21 | 72.03 | 44.07 | 73.96 | **62.24** | 17.24 | 65.00 | **58.54** | **64.24** |

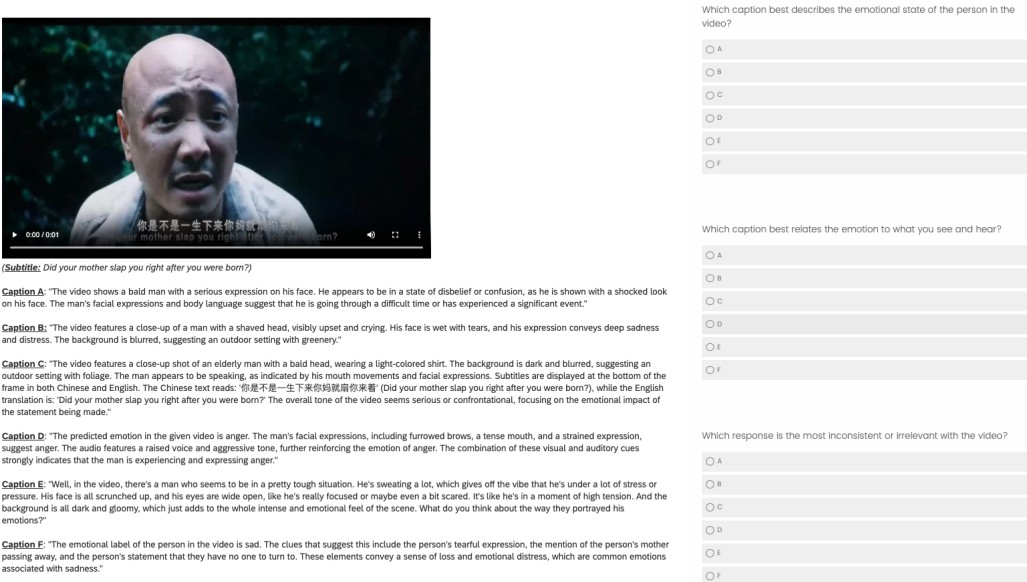

Figure 10: **User Evaluation using Qualtrics**. *(Left)* We show anonymized model responses for a given video to the user as different captions. *(Right)* We ask multiple questions to the user to select the best-suited caption for each question. Questions check the captions for their quality of emotion description, association of emotions with audiovisual cues, and presence of inconsistencies (hallucinations).

Table 16: User evaluation on EMER dataset.

| Model | Emot.↑ | Assoc.↑ | Incons.↓ |
|---|---|---|---|
| VideoLLaMA 2 | 9.82% | 0.75% | 15.38% |
| OLA | 9.36% | 7.46% | 5.58% |
| VITA 1.5 | 11.60% | 17.25% | 6.04% |
| Qwen 2.5 Omni | 10.75% | 18.57% | 10.13% |
| EmotionLLaMA | 1.89% | 11.53% | 68.61% |
| Our + AVEm-DPO | 54.74% | 43.35% | 4.67% |

### E.3 EMOTION RECOGNITION RESULTS - EXPANDED

Table 15 (expanded from Table 2) shows the results on DFEW (Jiang et al., 2020) emotion recognition benchmark over different emotion classes. Note that both our base model and AVEm-DPO trained model achieve the best and second-best results in terms of unweighted and weighted average recalls over all the emotion classes. Moreover, Table 15 shows that the proposed method ensures fair performance over all the emotion categories, unlike baselines, which perform too well on some classes and too poorly on the others.

### E.4 USER EVALUATION

We perform a user study on 40 participants recruited through Prolific (Prolific) and create a user survey using Qualtrics (Qualtrics) as shown in Fig. 10. We randomly sample videos from EMER (Lian et al., 2023b) dataset and display anonymized model generations as captions to the user along with the video. Then we ask the users to pick the most suited caption over different criteria – (i) best caption describing the emotional state of the person, (ii) best caption associating the emotion with audiovisual cues, and (iii) worst caption with the most inconsistencies with the video (to test model hallucinations). Table 16 (duplicate of Table 4) reports the average percent of times each model is selected for the mentioned three criteria. The participants selected our model the most number of times as the best model for emotion description and association of audiovisual cues for emotion. Moreover, our model was chosen the least number of times for inconsistent audiovisual information present in the caption.

Table 18: Performance variation over various choices of rejected response. $y_l^{irr}$: response completely irrelevant to the audiovisual content and emotion, $y_l^{er}$: response mentions hallucinated cues that generally co-occur with given emotion, $y_l^{vr}$: response associates audiovisual cues in the input incorrectly with emotion.

| $y_l^1$ | $y_l^2$ | **Basic** | **Agree.** | **Stress** | **Spur.** | **Hall.** |
|---------|---------|-----------|------------|------------|-----------|-----------|
| **Our base** | | 77.3 | 34.6 | 55.1 | 47.3 | 39.2 |
| $y_l^{irr}$ | - | 82.4 | 56.7 | 81.4 | 85.1 | 88.9 |
| $y_l^{er}$ | - | 84.0 | 58.3 | 86.0 | 88.5 | 97.9 |
| $y_l^{vr}$ | - | 83.2 | 58.0 | 85.3 | 91.6 | 90.9 |
| $y_l^{er}$ | $y_l^{irr}$ | 83.4 | 57.6 | 85.8 | 88.2 | 97.8 |
| $y_l^{vr}$ | $y_l^{irr}$ | 83.1 | 57.3 | 84.9 | 90.3 | 90.8 |
| $y_l^{er}$ | $y_l^{vr}$ | 85.2 | 60.1 | 87.8 | 92.7 | 97.6 |

## E.5 MODALITY PREFERENCE ABLATION

Table 17 shows AVEm-DPO's performance for different choices of multimodal preferences. We perform experiment using random tensor, random video, $(a_l, v_l)$ infused with diffusion noise similar to VCD (Leng et al., 2024) and an audiovisual input with different emotion than $(a_w, v_w)$ as the possible choices for $(a_l, v_l)$ and show that using a different emotion video leads to the best results. Moreover, we also show the effect of changing both $(a_w, v_w)$ vs. changing based on the input prompt ($a_w$ for audio reasoning, $v_w$ for visual reasoning and both for other tasks), justifying the effectiveness of prompt-based modality preference.

Table 17: Performance variation over various choices of rejected multimodal input. **Change** denotes which among $(a_w, v_w)$ should be changed to create $(a_l, v_l)$.

| Choice of $a_l/v_l$ | Change | Basic | Agree. | Stress |
|---------------------|--------|-------|--------|--------|
| Random tensor | Both $a_l, v_l$ | 81.9 | 56.1 | 80.1 |
| | Prompt-based | 83.0 | 56.0 | 81.6 |
| Random video | Both $a_l, v_l$ | 81.8 | 58.2 | 80.3 |
| | Prompt-based | 83.6 | 58.2 | 82.1 |
| Diffuse $(a_w, v_w)$ | Both $a_l, v_l$ | 82.7 | 58.5 | 80.9 |
| | Prompt-based | 84.6 | 59.4 | 86.7 |
| Diff. emotion | Both $a_l, v_l$ | 83.9 | 60.1 | 81.3 |
| | Prompt-based | 85.2 | 60.0 | 87.8 |

## E.6 RESPONSE PREFERENCE ABLATION

Table 18 shows the variation of performance over different tasks of *EmoReAlM* for different choices of rejected responses. There are three types of rejected responses that we test on – (i) $y_l^{vr}$ is video-relevant response that contains audiovisual cue present in the video, but it does not associate with the emotion, (ii) $y_l^{er}$ is emotion-relevant response that correctly associates with the emotion displayed in the video but with audiovisual cues that are hallucinated (not present in the video), and (iii) $y_l^{irr}$ is completely irrelevant to the given video and emotion (similar to that present in Huang et al. (2025b)). $y_l^1$ and $y_l^2$ in Table 18 denote the first and second rejected responses for preference tuning in Eq. (10).

We can see that our choice of using $y_l^{vr}$ and $y_l^{er}$ in Eq. (10) for AVEm-DPO results in the best performance of the model across all tasks. We also perform experiments using a single rejected response (Eq. (8)), and we can see that using $y_l^{er}$ and $y_l^{vr}$ individually results in improvement over the base, specifically for the *Spurious Cue-Emotion Association* and *Emotion-relevant Cue Hallucination* subtasks, respectively. Moreover, similar to Vista-DPO (Huang et al., 2025b), we perform an experiment using $y_l^{irr}$ as the second rejected response, which results in the same or worse performance than using $y_l^{vr}$ and $y_l^{er}$ alone. When using $y_l^{irr}$ as the second rejected response, we set $\beta_{irr} = 0.3$ following Huang et al. (2025b).

## E.7 SENSITIVITY TO HYPERPARAMETERS

Fig. 11 shows AVEm-DPO's accuracy on different subtasks of *EmoReAlM* on varying the hyperparameters $\beta_{vr}/\beta_{er}$ in Eq. (6). We can observe that while spurious cue-emotion associations mitigate on increasing $\beta_{vr}$, model performance on hallucinated cue samples improves on increasing $\beta_{er}$. For text-prior debiasing (TPD), we can see that performance on hallucinated cue samples significantly improves even with $\gamma_{TPD} = 0.1$ and gets saturated at $\gamma_{TPD} > 0.2$. Finally, increasing the strength of PMP using $\lambda_{av}$ (Eq. (9)) improves performance but it gets saturated at $\lambda_{av} > 1.0$.

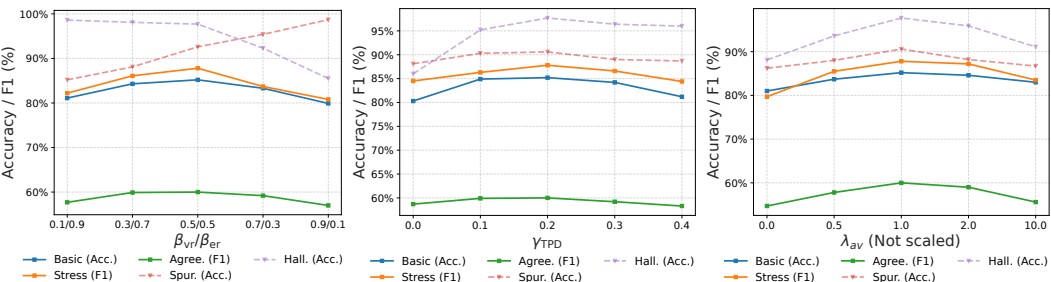

Figure 11: Sensitivity of performance to $\beta_{vr}/\beta_{er}$, $\gamma_{\text{TPD}}$ and $\lambda_{av}$.

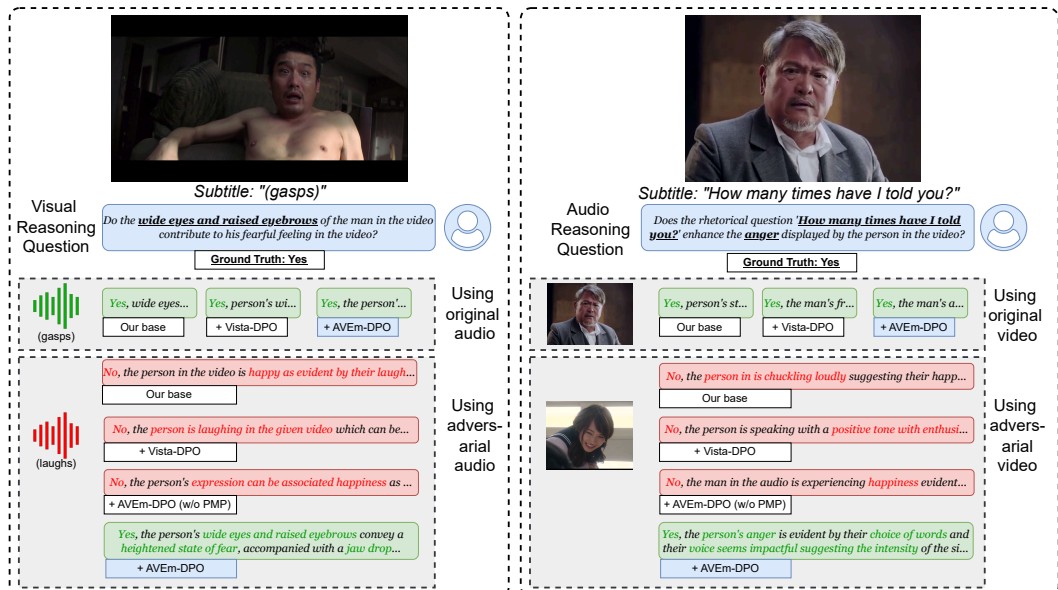

Figure 12: *(Left)* For a visual reasoning question, we compare the model responses on using the original video with the original audio and an adversarial audio as input. We can observe that Vista-DPO and even AVEm-DPO without prompt-based modality preference (PMP) struggle in the adversarial settings; however, AVEm-DPO produces the desired response. *(Right)* We perform a similar experiment to show the visual reasoning robustness of AVEm-DPO.

### E.8 ATTENTION REDISTRIBUTION AFTER PREFERENCE OPTIMIZATION.

As described in Section 5.2, to analyze the effect of preference optimization on attention, we plot the distribution of aggregate multimodal input attention over audio and visual tokens averaged over all attention heads for different tasks in *EmoReAlM* in Fig. 15 (*left two plots*). For reasoning tasks, we can observe that the attention over relevant modality increases after AVEm-DPO. For the *Modality Agreement* task, the attention is redistributed in a way that there is a fair distribution of attention between both modalities to ensure reliable predictions.

To show the effect of text-prior debiasing Section 4.2, we plot the percentage of total attention (averaged over attention heads) over multimodal input tokens (audio and visual combined) and observe that AVEm-DPO increases the attention over multimodal tokens by significant margins (Fig. 15 – *right*). This shows that AVEm-DPO training ensures that the model attends to the relevant audiovisual tokens for generating the response rather than relying only on the input text prompt.

### E.9 REASONING WITH ADVERSARIAL MODALITY INPUTS

To test the robustness of AVEm-DPO against cross-modality hallucinations, we conduct an adversarial test by replacing the audio in a visual reasoning task to see if the model's response stays

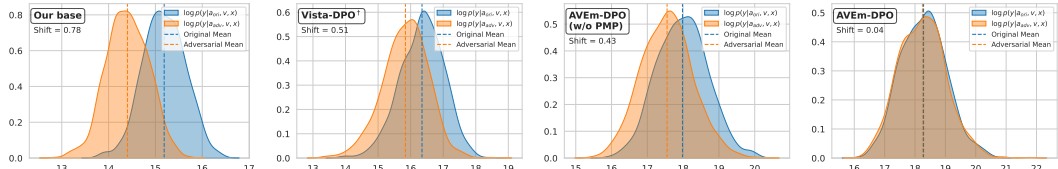

Figure 13: **Adversarial Audio Reasoning Testing**. For samples related to *audio* reasoning in the EmoReAlM benchmark (*Emotion Reasoning-Basic* and *Emotion Reasoning - Stress Test*), we plot the Kernel Density Estimation (KDE) shift in log likelihoods of the correct answer when the irrelevant video modality input ($v_{ori}$) is replaced with a random video as adversary ($v_{adv}$). AVEm-DPO is least affected by the addition of an adversary in the irrelevant modality (i.e., video).

Figure 14: **Adversarial Visual Reasoning Testing**. Similar to Fig. 13, for samples related to *visual* reasoning in the EmoReAlM benchmark (*Emotion Reasoning-Basic* and *Emotion Reasoning - Stress Test*), we plot the Kernel Density Estimation (KDE) shift in log likelihoods of the correct answer when the irrelevant audio modality input ($a_{ori}$) is replaced with a random video as adversary ($a_{adv}$). AVEm-DPO is least affected by the addition of an adversary in the irrelevant modality (i.e., audio).

the same. As shown in Fig. 12, changing the prompt-irrelevant modality does not change the response of AVEm-DPO, showing its adversarial robustness. It is interesting to note that removing the prompt-based modality preference (PMP) from AVEm-DPO results in wrong predictions, showing its efficacy. To quantitatively show the effect of AVEm-DPO with PMP, we perform adversarial testing using *Emotion Reasoning-Basic* and *Emotion Reasoning - Stress Test* samples in *EmoReAlM*. For testing the robustness of AVEm-DPO for audio reasoning (Fig. 13), we compute the shift of log likelihoods of the correct response when the prompt-irrelevant video modality is replaced with an adversary (i.e., some random video). We use Kernel Density Estimation (KDE) to estimate the shift in the distributions. We can see that AVEm-DPO is robust to adversaries in the prompt-irrelevant modality. Moreover, removing PMP from AVEm-DPO significantly increases the shift between the original and adversarial distributions. Fig. 14 shows similar plots for the tasks related to visual reasoning.

### E.10 EFFECT OF INDIVIDUAL MODALITIES FOR EMOTION PREDICTION

To show the effect of individual modalities for emotion recognition, Table 19 reports the performance on using only the video, only the audio and using audiovisual inputs from the RAVDESS (Livingstone & Russo, 2018) dataset for emotion prediction. We can observe that using the individual modalities for emotion prediction leads to a reduced performance, indicating that using the audiovisual inputs for emotion prediction is indeed helpful compared to using a single modality.

Table 19: Performance of different models for emotion prediction using audiovisual, video-only, and audio-only inputs from the RAVDESS (Livingstone & Russo, 2018) dataset.

| Model | Audiovisual | | Video-only | | Audio-only | |
|---|---|---|---|---|---|---|
| | UAR | WAR | UAR | WAR | UAR | WAR |
| VideoLLaMA 2 | 41.81 | 31.62 | 36.12 | 32.41 | 30.44 | 27.56 |
| Qwen 2.5 Omni | 32.88 | 28.05 | 29.38 | 27.67 | 28.56 | 25.55 |
| Our base | 53.59 | 53.01 | 41.27 | 40.98 | 38.18 | 37.74 |
| + AVEm-DPO | 58.66 | 55.48 | 46.13 | 46.31 | 44.05 | 39.67 |
| EmotionLLaMA⋆ | 52.59 | 48.12 | 41.27 | 39.56 | 38.57 | 37.27 |
| + AVEm-DPO | 56.21 | 51.03 | 46.10 | 43.09 | 40.54 | 37.04 |

Figure 15: Effect of AVEm-DPO on the distribution of attention over – (i) *(Left three plots)* video and audio tokens taken as a percentage over the total attention over all multimodal tokens for different subtasks in *EmoReAlM* and (ii) *(Right)* multimodal tokens as a percentage over the total input tokens (including text) for the entire *EmoReAlM*.

Moreover, we can observe that the performance using only the visual modality is better compared to using only audio, indicating the importance of visual modality for emotion prediction.

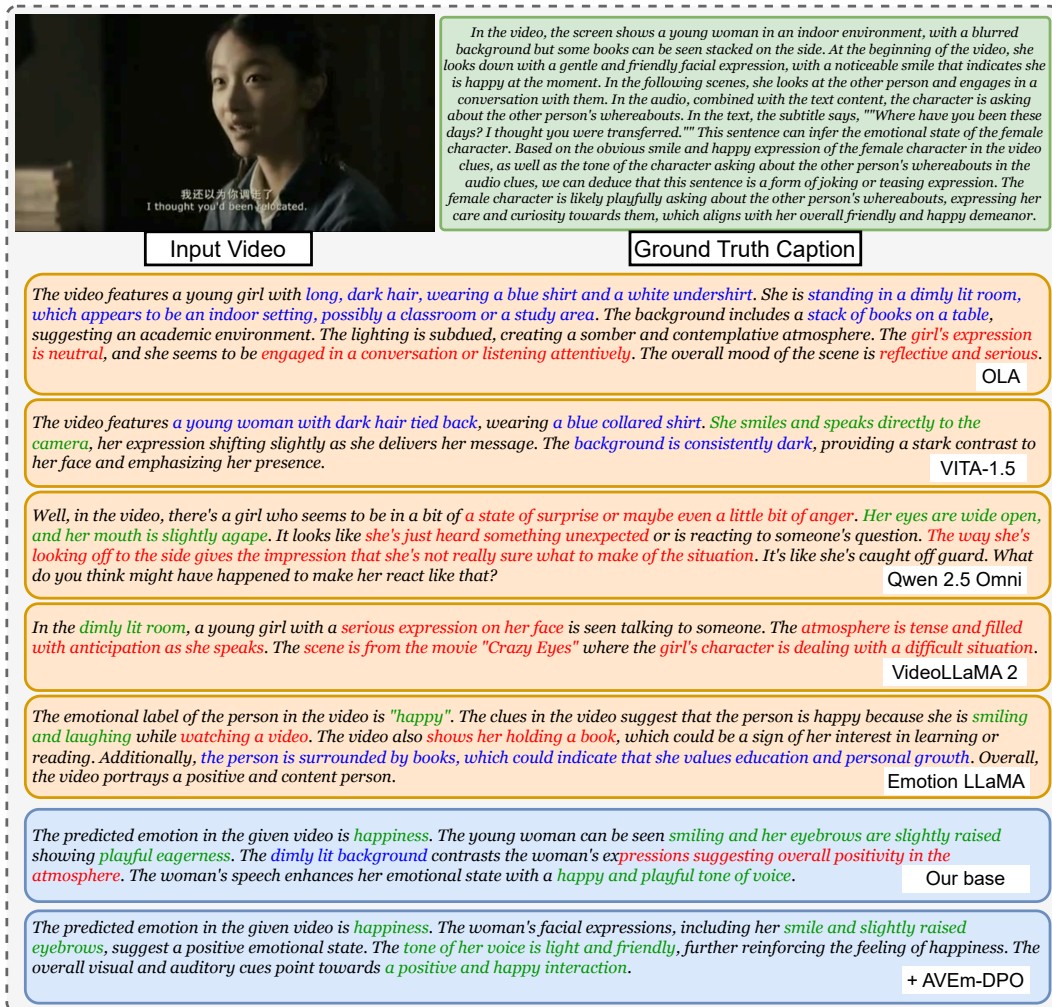

Figure 16: Comparison of baseline MLLMs with our base model trained with AVEm-DPO on a sample from EMER (Lian et al., 2023b). Correct audiovisual cues and emotion are in green, emotion-irrelevant cues are in blue, and hallucinated cues (and incorrect emotion) are present in red
.

# F  QUALITATIVE SAMPLES

**Emotion Descriptions on EMER.**   Figs. 16 and 17 shows samples from the EMER (Lian et al., 2023b) dataset and the output of different MLLM baselines on those samples using the prompt –
*"Describe the audiovisual content relevant to emotion in detail."*.   We can see that AVEM-DPO
leads to correct emotion descriptions and consistent audiovisual cues to reason for the emotions.
Moreover, compared to baselines, our method does not associate irrelevant and/or background information with emotions.

**EmoReAlM Sample Outputs.**   Fig. 18 shows the model responses for some samples in the *Emotion Reasoning - Stress Test* of *EmoReAlM* Benchmark.  We can notice that AVEm-DPO improves
the model responses in cases with spurious-emotion cue associations and emotion-cue hallucinations.

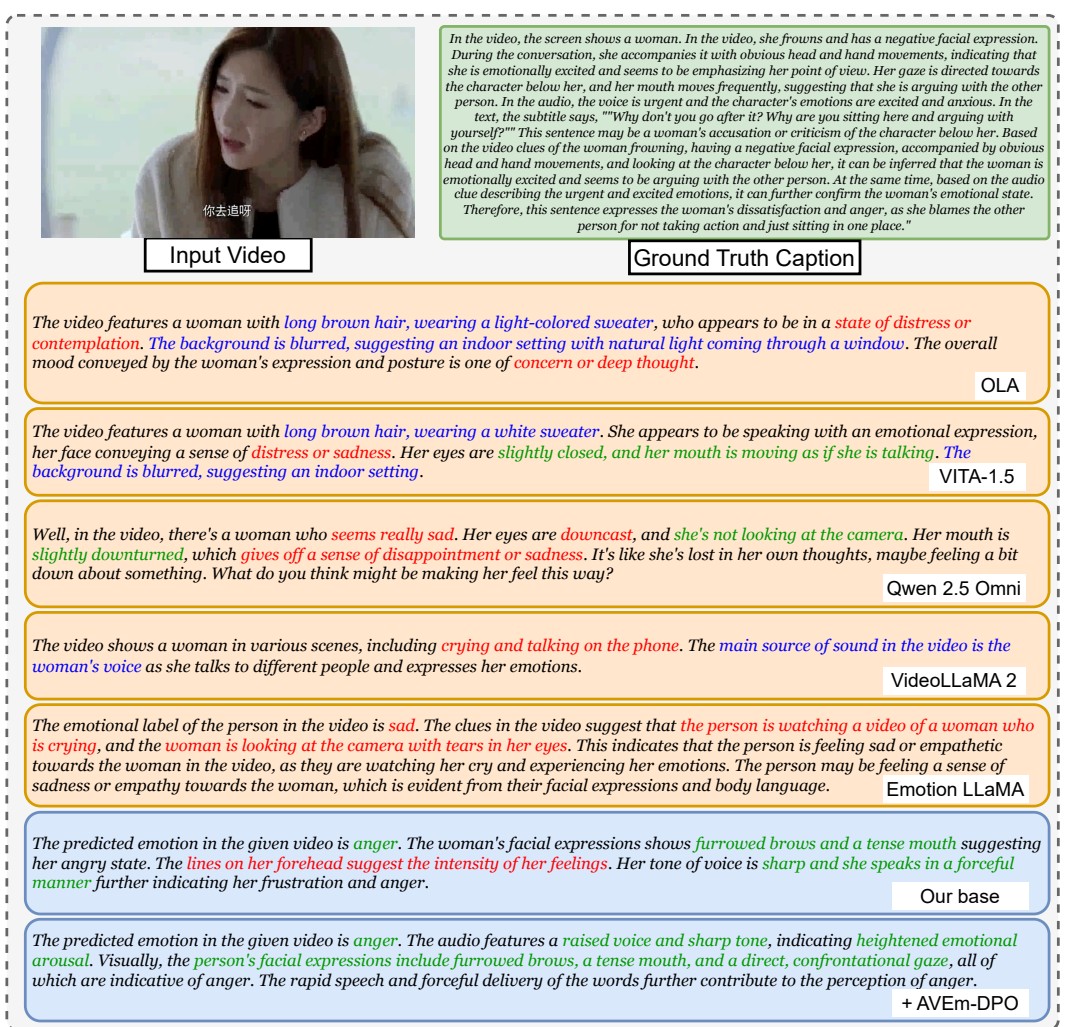

Figure 17: Comparison of baseline MLLMs with our base model trained with AVEm-DPO on a sample from EMER (Lian et al., 2023b). Correct audiovisual cues and emotion are in green, emotion-irrelevant cues are in blue, and hallucinated cues (and incorrect emotion) are present in red

.

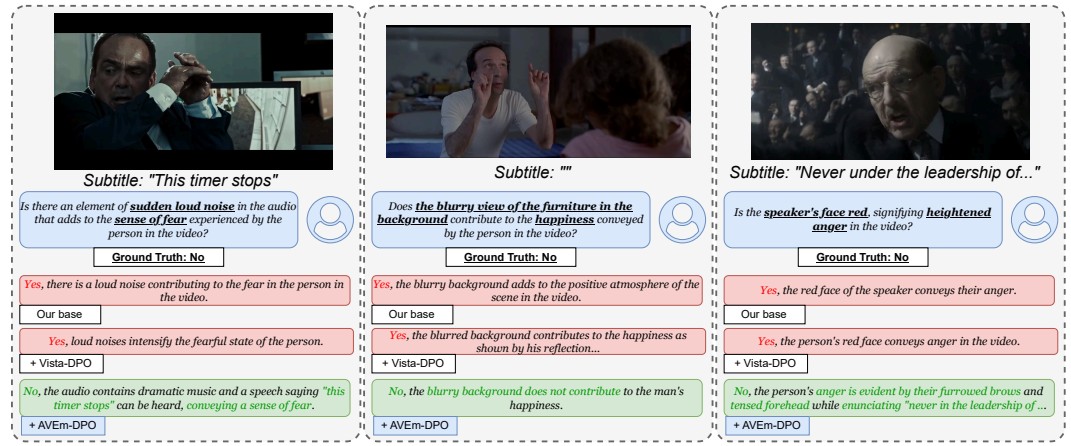

Figure 18: Qualitative examples comparing the output responses using different approaches for some samples present in the *Emotion Reasoning - Stress Test* of EmoReAlM benchmark.

## G   PROMPT POOL

You are an expert in audio captioning. Your task is to provide a detailed caption for the given audio, while covering as much information as possible.

Keep the following points in mind while generating the caption:
1. Describe the audio content such as transcript, speech, tone of voice, background noise, music, sound effects, etc.
2. Focus on the audio cues which can explain the emotional state of the video or the characters present in the video.
3. If the speech is in a language other than English, provide transcript in other language as well as English translation.
4. DO NOT STATE ANYTHING ABOUT THE VISUAL CONTENT OF THE VIDEO.

Return your response strictly in the following JSON format: {"detailed_caption": "... detailed caption about everything ...", "emotion_caption": "... detailed caption only about the tone of voice, speech content or any other detail which deals with emotion ..."}

Figure 19: **Audio caption prompt** – used to caption only the audio content from a video. Note that the audio is passed along with the prompt to GPT-4o-audio as a WAV file.

You are an expert in video captioning. Your task is to provide a detailed caption for the given video, while covering as much information as possible.

Only focus on the visual content and ignore the subtitle if it is present in the video.

Keep the following points in mind while generating the caption:
1. Describe the visual content such as facial expression of the character(s) in detail. Additionally, comment on the body language, gestures of the character(s) as well as the background or setting of the given video.
2. Focus on the visual cues which can explain the emotional state of the character(s) in the video and the video in general.
3. DO NOT STATE ANYTHING ABOUT THE SUBTITLE OR AUDIO CONTENT OF THE VIDEO.

Return your response strictly in the following JSON format: {"detailed_caption": "... detailed caption about everything ...", "emotion_caption": "... detailed caption only about the facial expressions, body language, gestures, or any aspect of the video which deals with emotion ..."}

Figure 20: **Video caption prompt** – used to caption only the visual content in a video. We blur the captions if they are already present in the video and explicitly ask the model to ignore them if they are present in the visual content.

You will be given an audio caption from a video and your task is to predict the emotion displayed just with the audio caption. Label can be one of the following: "happiness", "sadness", "anger", "fear", "disgust", "surprise", and "neutral".

Try to predict the closest emotion label based on the audio caption and do not return disclaimers or anything else. Focus on the audio transcript as well as the tone of voice and avoid predicting neutral unless absolutely necessary.

Return your response in the following JSON format - {"video_id": video_id, "emotion": "emotion_label"}.
"emotion_label" should be a single word, one of the following: "happiness", "sadness", "anger", "fear", "disgust", "surprise", and "neutral".

Video ID: *"{VIDEO ID}"*
Audio Caption: *"{AUDIO CAPTION}"*

Figure 21: **Audio emotion prediction prompt** – used to predict the emotion into one of the 7 basic categories from only the audio caption.

You will be given a video caption from a video and your task is to predict the emotion displayed just with the video caption. Label can be one of the following: "happiness", "sadness", "anger", "fear", "disgust", "surprise", and "neutral".

Try to predict the closest emotion label based on the video caption and do not return disclaimers or anything else.

Return your response in the following JSON format - {"video_id": video_id, "emotion": "emotion_label"}.
"emotion_label" should be a single word, one of the following: "happiness", "sadness", "anger", "fear", "disgust", "surprise", and "neutral".

Video ID: *"{VIDEO ID}"*
Video Caption: *"{VIDEO CAPTION}"*

Figure 22: **Video emotion prediction prompt** – used to predict the emotion into one of the 7 basic categories from only the video caption

You will be provided with an audio caption and an emotion label associated with a video.
Your task is to create high quality question-answer pairs based on the provided audio caption and emotion label. The questions should be asking about the audio content responsible for the emotion in the video.
The audio caption contains all the details about the emotional content of the audio and other context/music/background noise in the audio.

Keep the following points in mind while generating the question answer pairs:
1. The questions should be focussed on reasoning about the given emotion based on audio cues without any explicit mention of the displayed emotion. The question should not mention the given emotion label in any form.
2. Each question should have 4 choices (A, B, C, D), one of which should be the correct answer.
3. The incorrect choices can be either (i) plausible audio cues to explain the correct emotion not present in the audio (ii) audio cues present in the audio caption but do not contribute to the emotion displayed in the audio. For example, if the emotion is "sadness", the incorrect choices should be audio cues that can explain sadness but are not present in the audio caption.
4. All the choices should be of almost equal length.
5. For each question, provide the correct answer both in terms of the correct choice (A, B, C, D) and in the form of a text answer. The text answer should be the detailed version of the correct answer to the question, without any mention of the choices.
6. DO not frame questions that include phrases such as "audio caption" or "audio transcript" or "which of the following" or "what best suits", etc.

Example questions:
1. [Label-Happiness][Semantic Speech] How does the man's words display the emotion in the video? (A) The man says that they received a promotion at work (B) The man says that they are pregnant with their first child (C) The man says that they just won a lottery (D) The man says that they are going on a vacation.
2. [Label-Sadness][Paralinguistics] How does the character's tone of voice contribute to their emotional state? (A) The character is speaking in a low, sad tone (B) The character is crying with a shaky voice (C) The character is speaking in a monotone voice suggesting depression (D) The character is speaking in a high-pitched voice suggesting anxiety.

Return your response strictly in the following JSON format - {"video_id": video_id, "questions": [{"question": "Question text", "choices": ["(A) choice A", "(B) choice B"...], "answer": {"choice": "C", "text":"answer text"}, "category":"semantic_speech_reasoning/paralinguistic_speech_reasoning"}, ...]}
Return "ERROR" if you are unable to generate any question answer pairs. Also specify why you are unable to generate the question answer pairs.

Generate at least one question about semantic speech, and one about paralinguistic speech.
If the audio caption does not suggest the given emotion label, then do not generate any question answer pairs and return "ERROR".

===
Video ID: *"{VIDEO ID}"*
Emotion Label: *"{EMOTION}"*
Audio Caption: *"{AUDIO CAPTION}"*

Figure 23: **EmoReAlM Basic Reasoning Prompt - Audio** – used to generate questions which ask about the audio cues that suggest the emotion of the person in the video.

You will be provided with a video caption and an emotion label associated with the video.
Your task is to create high quality question-answer pairs based on the provided video caption and emotion label. The questions should be asking about the visual content responsible for the emotion in the video.
The video caption contains all the details about the emotional content of the video, including detailed visual content.

Keep the following points in mind while generating the question answer pairs:
1. The questions should be focussed on reasoning about emotion displayed in the video without any explicit mention of the displayed emotion. The question should not mention the given emotion label in any form.
2. Each question should have 4 choices (A, B, C, D), one of which should be the correct answer.
3. The incorrect choices can be either (i) plausible visual cues to explain the correct emotion not present in the video (ii) visual cues present in the video caption but do not contribute to the emotion displayed in the video. For example, if the emotion is "sadness", the incorrect choices should be visual cues that can explain sadness but are not present in the video caption.
4. All the choices should be of almost equal length.
5. For each question, provide the correct answer both in terms of the correct choice (A, B, C, D) and in the form of a text answer. The text answer should be the detailed version of the correct answer to the question, without any mention of the choices.
6. DO not frame questions that include phrases such as "video caption" or "video transcript" or "which of the following" or "what best suits", etc.

Example questions:
1. [Label-Happiness][Facial Expression] How does the man's facial expression contribute to the emotion displayed in the video? (A) The man is laughing with a big smile (B) The man smirks slightly with an implicit happiness in his eyes (C) The man bursts into laughter suggesting extreme joy (D) The man's eyes are filled with tears of joy.
2. [Label-Sadness][Body Language] How does the character's gesture contribute to their emotional state? (A) The character is slumped over with their head down suggesting melancholy (B) The character's posture suggests a lack of confidence and depression suggesting sadness (C) The character cries with their hands covering their face (D) The character is sitting with their arms crossed and looking down, suggesting sadness.

Return your response strictly in the following JSON format - {"video_id": video_id, "questions": [{"question": "Question text", "choices": ["(A) choice A", "(B) choice B"...], "answer": {"choice": "C", "text":"answer text"}, "category":"facial_expression_reasoning/body_language_reasoning"}, ...]}
Return "ERROR" if you are unable to generate any question answer pairs. Also specify why you are unable to generate the question answer pairs.

Generate at least one question about facial expression, and one about body language.
If the video caption does not suggest the given emotion label, then do not generate any question answer pairs and return "ERROR".

===
Video ID: *"{VIDEO ID}"*
Emotion Label: *"{EMOTION}"*
Video Caption: *"{VIDEO CAPTION}"*

Figure 24: **EmoReAlM Basic Reasoning Prompt - Visual** – used to generate questions which ask about the visual cues that suggest the emotion of the person in the video.

You are an expert in audio-visual emotion understanding and analysis. You will be given audio captions and video captions for an audio-visual content, along with the manually annotated emotion label out of "happiness", "sadness", "anger", "fear", "disgust", "surprise", and "neutral".
Your task is to analyze the audio and video captions and denote whether the audio and video modalities agree with each other in conveying the emotion label.
Finally, you have to generate question answer pairs asking about the modality agreement of the audio-video content in conveying the emotion. You should frame questions about the video and not the captions.

Do not generate any question answer pairs if neither the audio nor the video content convey the emotion label.

Following are a few examples of the questions that you can ask. DO NOT ASK THE SAME QUESTIONS AS GIVEN BELOW, BUT GENERATE SIMILAR QUESTIONS:
1. Are the audio and video modalities in agreement with each other in conveying the emotion of the video? (A) Yes (B) No -- Modality agreement
2. Are the visual and audio cues in agreement to convey the sadness in the video? (A) Yes (B) No -- Modality agreement

Return your response strictly in the following JSON format: {"video_id": video_id, "questions": [{"question": "Question text", "choices": ["(A) choice A", "(B) choice B"...], "answer": {"choice": "C", "text":"answer text"}, "category":"modality_agreement"}, ...]}
In the "answer_text" field, give a detailed explanation of the correct answer to the question, without any mention of the choices.

RETURN "ERROR" if you are unable to generate any question answer pairs. Also specify why you are unable to generate the question answer pairs.

===
Video ID: *"{VIDEO ID}"*
Emotion Label: *"{EMOTION}"*
Audio Caption: *"{AUDIO CAPTION}"*
Video Caption: *"{VIDEO CAPTION}"*

Figure 25: **EmoReAlM Modality Agreement Prompt** – used to generate questions which ask whether the audio and video in the audiovisual input align with each other to express the emotion in the video.

You will be provided with an audio caption for a video and an emotion label associated with the video. The audio caption will contain some information related to the emotion label.
Your task is to generate a question of the format - "Does the {...some audio cue...} contribute to the {...emotion...} experienced by the person in the video?" but not in the same words.
The audio cue mentioned in the question should be an audio cue (e.g. tone of voice or choice of words or something else) that is present in the given audio caption, and supports the emotion label given to you.

Return your response strictly in the following JSON format - {"video_id": video_id, "questions": [{"question": "Question text", "choices": ["(A) Yes", "(B) No"], "answer": {"choice": "A", "text":"explanation for your answer"}, "category":"audio_driven_audio_no_hallucination"}, ...]}

Only generate one question for the given inputs. Return the string "ERROR" if you are unable to generate any question or for something else.
Provide your reasoning in the "answer_text" field of the answer in terms of the video and not the caption. Your answer should always be "A" since the audio cue supports the emotion. Do not frame your answers in terms of captions, but rather in terms of video.
===
Video ID: *"{VIDEO ID}"*
Emotion Label: *"{EMOTION}"*
Audio Caption: *"{AUDIO CAPTION}"*

Figure 26: **EmoReAlM Stress Test Prompt - Audio - No Hallucination** – used to generate questions where the audio cue mentioned in the question is present in the audiovisual input and supports the emotion of the person in the video.

You will be provided with an audio caption for a video and an emotion label associated with the video.

Your task is to generate a question of the format - "Does the {...some audio cue...} contribute to the {...emotion...} experienced by the person in the video?" but not in the same words.
The audio cue mentioned in the question should be an audio cue (some auditory element irrelevant to emotion) that is present in the given audio caption, but does not support the emotion in any way remotely.

Return your response strictly in the following JSON format - {"video_id": video_id, "questions": [{"question": "Question text", "choices": ["(A) Yes", "(B) No"], "answer": {"choice": "B", "text":"explanation for your answer"}, "category":"audio_driven_audio_hallucination_audio_relevant"}, ...]}

Only generate one question for the given inputs. Return the string "ERROR" if you are unable to generate any question because all the audio cues in the audio caption align with the given emotion or for something else.
Provide your reasoning in the "answer_text" field of the answer in terms of the video and not the caption. Your answer should always be "B" since the audio cue does not support the emotion in the video. Do not frame your answers in terms of captions, but rather in terms of video.
===
Video ID: *"{VIDEO ID}"*
Emotion Label: *"{EMOTION}"*
Audio Caption: *"{AUDIO CAPTION}"*

Figure 27: **EmoReAlM Stress Test Prompt - Audio - Spurious Associations** – used to generate questions where the audio cue mentioned in the question is present in the audiovisual input and but it is spuriously related to the emotion of the person in the video.

You will be provided with an audio caption for a video and an emotion label associated with the video.

Your task is to generate a question of the format - "Does the {...some audio cue...} contribute to the {...emotion...} experienced by the person in the video?" but not in the same words.
The audio cue mentioned in the question should be an audio cue (preferably some words or phrases or tone of voice or some other auditory element) that is associated with the given emotion generally, but is NOT present in the audio caption.

Return your response strictly in the following JSON format - {"video_id": video_id, "questions": [{"question": "Question text", "choices": ["(A) Yes", "(B) No"], "answer": {"choice": "B", "text":"explanation for your answer"}, "category":"audio_driven_audio_hallucination_emotion_relevant"}, ...]}

Only generate one question for the given inputs. Return the string "ERROR" if you are unable to generate any question because all the audio cues generally associated with the given emotion are present in the audio caption or for something else.
Provide your reasoning in the "answer_text" field of the answer in terms of the video and not the caption. Your answer should always be "B" since the audio cue does not support the emotion in the video. Do not frame your answers in terms of captions, but rather in terms of video.
===
Video ID: *"{VIDEO ID}"*
Emotion Label: *"{EMOTION}"*
Audio Caption: *"{AUDIO CAPTION}"*

Figure 28: **EmoReAlM Stress Test Prompt - Audio - Hallucination** – used to generate questions where the audio cue mentioned in the question is hallucinated (not present in the audiovisual input) and but it usually explains the emotion experienced by the person in the video.

You will be provided with a video caption and an emotion label associated with the video. The video caption will contain some information related to the emotion label.
Your task is to generate a question of the format - "Does the {...some visual cue...} contribute to the {...emotion...} experienced by the person in the video?" but not in the same words.
The visual cue mentioned in the question should be a visual cue (either facial expression or body language or something else) that is present in the given video caption and it supports the emotion label given to you.

Return your response strictly in the following JSON format - {"video_id": video_id, "questions": [{"question": "Question text", "choices": ["(A) Yes", "(B) No"], "answer": {"choice": "A", "text":"explanation for your answer"}, "category":"video_driven_visual_no_hallucination"}, ...]}

Only generate one question for the given inputs. Return the string "ERROR" if you are unable to generate any question or for something else.
Provide your reasoning in the "answer_text" field of the answer in terms of the video and not the caption. Your answer should always be "A" since the visual cue supports the emotion. Do not frame your answers in terms of captions, but rather in terms of video.
===
Video ID: *"{VIDEO ID}"*
Emotion Label: *"{EMOTION}"*
Video Caption: *"{VIDEO CAPTION}"*

Figure 29: **EmoReAlM Stress Test Prompt - Video - No Hallucination** – used to generate questions where the visual cue mentioned in the question is present in the audiovisual input and supports the emotion of the person in the video.

You will be provided with a video caption and an emotion label associated with the video.

Your task is to generate a question of the format - "Does the {...some visual cue...} contribute to the {...emotion...} experienced by the person in the video?" but not in the same words.
The visual cue mentioned in the question should be a visual cue (something unrelated and irrelevant to emotion) that is present in the given video caption, but does not support the emotion in any way remotely.

Return your response strictly in the following JSON format - {"video_id": video_id, "questions": [{"question": "Question text", "choices": ["(A) Yes", "(B) No"], "answer": {"choice": "B", "text":"explanation for your answer"}, "category":"audio_driven_visual_hallucination_video_relevant"}, ...]}

Only generate one question for the given inputs. Return the string "ERROR" if you are unable to generate any question because the visual caption align with the given emotion or for something else.
Provide your reasoning in the "answer_text" field of the answer in terms of the video and not the caption. Your answer should always be "B" since the visual cue does not support the emotion. Do not frame your answers in terms of captions, but rather in terms of video.
===
Video ID: *"{VIDEO ID}"*
Emotion Label: *"{EMOTION}"*
Video Caption: *"{VIDEO CAPTION}"*

Figure 30: **EmoReAlM Stress Test Prompt - Video - Spurious Associations** – used to generate questions where the visual cue mentioned in the question is present in the audiovisual input and but it is spuriously related to the emotion of the person in the video.

You will be provided with a video caption and an emotion label associated with the video.

Your task is to generate a question of the format - "Does the {...some visual cue...} contribute to the {...emotion...} experienced by the person in the video?" but not in the same words.
The visual cue mentioned in the question should be a visual cue (preferebly a facial expression) that is associated with the given emotion generally, but is NOT present in the video caption.

Return your response strictly in the following JSON format - {"video_id": video_id, "questions": [{"question": "Question text", "choices": ["(A) Yes", "(B) No"], "answer": {"choice": "B", "text":"explanation for your answer"}, "category":"audio_driven_visual_hallucination_emotion_relevant"}, ...]}

Only generate one question for the given inputs. Return "ERROR" if you are unable to generate any question. Also specify why you are unable to generate the question.
Provide your reasoning in the "answer_text" field of the answer in terms of the video and not the caption. Your answer should always be "B" since the visual cue is not present in the video caption. Do not frame your answers in terms of captions, but rather in terms of video.
===
Video ID: *"{VIDEO ID}"*
Emotion Label: *"{EMOTION}"*
Video Caption: *"{VIDEO CAPTION}"*

Figure 31: **EmoReAlM Stress Test Prompt - Video - Hallucination** – used to generate questions where the visual cue mentioned in the question is hallucinated (not present in the audiovisual input) and but it usually explains the emotion experienced by the person in the video.

You are an intelligent assistant and good with logical reasoning. Your task is to guess the answer to the given question about some video, without access to the video itself.
The question will be about the emotional content of the video focussing on the audio-visual content of the video.
Keep the following points in mind while guessing the answer:
1. Your guess should not be random. It should be based on some hint provided in the question and the answer choices.
2. Do not provide an answer if you can not guess the answer based just on the text of the question and the answer choices.

Following are some examples of question answer pairs and their guesses:
Question 1: Why does the person in the video look sad? (A) They just lost a game. (B) Their grandma gave them a gift. (C) Their father gave them a hearty hug (D) They ate something that they are very fond of.
Guess Answer 1: (A)
Reason 1: Only one of the options is related to the emotion "sad" and the other options are about positive emotions.
Question 2: What is the emotion of the person in the video? (A) happy (B) sad (C) angry (D) surprised
Guess Answer 2: None
Reason 2: The question is about identifying the emotion of the person in the video, but we do not have access to the video to guess the answer.
Question 3: Are the audio video modalities in agreement with each other in conveying the emotion of the video? (A) Yes (B) No
Guess Answer 3: None
Reason 3: The question is about the agreement between audio and video modalities, but we do not have access to the video or audio to guess the answer.

Return your answer in the following JSON format: {"sample_id": sample_id, "guess": "A/B/C/D/None", "reason": "Reason for the guess"}

Following is the question to evaluate:
SAMPLE_ID: *"{SAMPLE_ID}"*
QUESTION: *"{QUESTION_WITH_CHOICES}"*

Figure 32: **Text Only Guess Prompt** – used to prompt GPT-4o, Gemini 2.5 and Qwen 2.5 to predict the answer to the generated QA samples using only the question text and answer choices to eliminate responses which can be answered just with the text.

You will be provided with a MCQ question, choices and answer related to reasoning about the emotion of a video based on the audio content.
Your task is to reformulate the answer choices to make the question more difficult.
You will also be provided with the audio captions of the video which the question is about along with the ground truth (GT) emotion label.

Specifically, you have to do the following checks:
1. If the question is not about reasoning the emotional state of the person based on the audio content or choice of words, then return "ERROR... {reason for your response}".
2. For the correct choice, rephrase the answer choice to make sure that it reasons for the emotion given as the GT emotion label and it is based on the speech content specified in the audio caption.
3. The incorrect choices should be reformulated to in a way so that they follow the same tone and length as the correct choice and they should attribute to the GT emotion label, but they differ from the correct choice in the following ways:
    a. [audio_relevant_emotion_irrelevant] One of the incorrect choice should pick a different audio cue (e.g. tone of voice, background sounds/music) from the audio caption and attribute it to the GT emotion label.
    b. [emotion_relevant_audio_irrelevant] Next incorrect choice should pick an audio cue that is generally associated with the GT emotion label but is NOT present in the audio caption. Do not mention in the choice that the audio cue is not present in the audio caption, just rephrase it as if it is present.
    c. [all_irrelevant] The last incorrect choice should be some audio cue that is not present in the audio caption, but it should sound similar to the correct choice and it should attribute to the GT emotion label.
4. Do not explicitly mention that you are given the emotion label. Do not mention captions in the question or the answer choices.
5. Phrase all the responses in a way that they explain the given GT emotion label and answer the question.

Return your response strictly in the following JSON format - {"video_id": video_id, "question": "Question text", "choices": {"correct": "correct choice rephrased", "audio_relevant_emotion_irrelevant": "incorrect choice a.", "emotion_relevant_audio_irrelevant": "incorrect choice b.", "all_irrelevant": "incorrect choice c."}}.
Return "ERROR" if you are unable to reformulate the question and answer choices. Also specify why you are unable to reformulate the question and answer choices.
===
Video ID: *"{VIDEO_ID}"*
Emotion label: *"{EMOTION_LABEL}"*
Audio Caption: *"{AUDIO_CAPTION}"*
Question: *"{QUESTION}"*
Choices: *"{CHOICES}"*
Answer: *"{ANSWER}"*

Figure 33: **Preference Data Generation Prompt - Audio Reasoning** – used to generate rejected responses for a generated question-answer pair related to audio reasoning tasks.

You will be provided with a MCQ question, choices and answer related to reasoning about the emotion of a video based on the visual content of a video.
Your task is to reformulate the answer choices to make the question more difficult.
You will also be provided with the video caption of the video which the question is about along with the ground truth (GT) emotion label.

Specifically, you have to do the following checks:
1. If the question is not about reasoning the emotional state of the person based on the visual content, then return "ERROR... {reason for your response}".
2. For the correct choice, rephrase the answer choice to make sure that it reasons for the emotion given as the GT emotion label and it is based on the visual content specified in the video caption.
3. The incorrect choices should be reformulated to in a way so that they follow the same tone and length as the correct choice and they should attribute to the GT emotion label, but they differ from the correct choice in the following ways:
    a. [video_relevant_emotion_irrelevant] One of the incorrect choice should pick a different visual cue (e.g. background information, color of attire, body language, etc.) from the video caption and attribute it to the GT emotion label.
    b. [emotion_relevant_video_irrelevant] Next incorrect choice should pick a visual cue that is usually associated with the GT emotion label, but it is not present in the video caption. Do not mention in the choice that the visual cue is not present in the video caption, just rephrase it as if it is present.
    c. [all_irrelevant] The last incorrect choice should be a visual cue that is NOT present in the video caption and does NOT explain the emotion in general and attribute it to the GT emotion label. Do not mention in the choice that the visual cue is not present in the video caption, just rephrase it as if it is present.
4. Do not explicitly mention that you are given the emotion label. Do not mention captions in the question or the answer choices.
5. Phrase all the responses in a way that they explain the given GT emotion label and answer the question.

Return your response strictly in the following JSON format - {"video_id": video_id, "question": "Question text", "choices": {"correct": "correct choice rephrased", "video_relevant_emotion_irrelevant": "incorrect choice a.", "emotion_relevant_video_irrelevant": "incorrect choice b.", "all_irrelevant": "incorrect choice c."}}.
Return "ERROR" if you are unable to reformulate the question and answer choices. Also specify why you are unable to reformulate the question and answer choices.
===
Video ID: *"{VIDEO_ID}"*
Emotion label: *"{EMOTION_LABEL}"*
Video Caption: *"{VIDEO_CAPTION}"*
Question: *"{QUESTION}"*
Choices: *"{CHOICES}"*
Answer: *"{ANSWER}"*

Figure 34: **Preference Data Generation Prompt - Visual Reasoning** – used to generate rejected responses for a generated question-answer pair related to visual reasoning tasks.

You will be provided with a MCQ question, choices and answer related to whether the audio and video modalities agree in conveying the same emotion for the character in the video or not.
Your task is to create new answer choices to make the question more difficult.
You will also be provided with the video and audio captions of the video which the question is about along with the ground truth (GT) emotion label.

Specifically, you have to do the following checks:
1. If the question is not about modality agreement for the emotional state of the person, then return "ERROR... {reason for your response}".
2. For the correct choice between "Yes" or "No", rephrase the answer choice to make sure that it explains either yes or no based on the GT emotion label and the audio and video captions.
3. I want you to generate the following incorrect choices in a way so that they follow the same tone and length as the correct choice and they should reason for the opposite answer (opposite of correct answer in 2), but they differ from the correct choice in the following ways:
    a. [video_relevant_emotion_irrelevant] One of the incorrect choice should pick a different audio or visual cue (e.g. tone of voice, background colour of wall, outfit colour, background sounds/music) from the audio and video caption and attribute it to the opposite answer.
    b. [emotion_relevant_video_irrelevant] Next incorrect choice should pick audio and visual cues that support the opposite answer (complementary cues if opposite answer is "Yes", else contradictory cues if opposite answer is "No") but are not present in the given audio or video caption. Do not mention in the choice that the video/audio cue is not present in the video/audio caption, just rephrase it as if it is present.
    c. [all_irrelevant] Final incorrect choice should be completely irrelevant to the question and should talk about some audio visual cues that are not even present in the given captions.
4. Do not explicitly mention that you are given the emotion label. Do not mention captions in the question or the answer choices.
5. Phrase all the responses in a way that they explain whether the audio and video modalities agree or disagree in conveying the same emotion for the character in the video or not.
6. Start your answer with either "Yes, ..." or "No, ...".
7. If the question is not about the emotional state of the person or character, rephrase it in a way that it is about the emotional state/mood/feeling of the person. It is important that the question is about emotion of a person and not general emotion.

Return your response strictly in the following JSON format - {"video_id": video_id, "question": "Question text", "choices": {"correct": "correct choice rephrased", "video_relevant_emotion_irrelevant": "incorrect choice a.", "emotion_relevant_video_irrelevant": "incorrect choice b."}}.
Return "ERROR" if you are unable to reformulate the question and answer choices. Also specify why you are unable to reformulate the question and answer choices.
===
Video ID: *"{VIDEO_ID}"*
Emotion label: *"{EMOTION_LABEL}"*
Audio Caption: *"{AUDIO_CAPTION}"*
Video Caption: *"{VIDEO_CAPTION}"*
Question: *"{QUESTION}"*
Choices: *"{CHOICES}"*
Answer: *"{ANSWER}"*

Figure 35: **Preference Data Generation Prompt - Modality Agreement** – used to generate rejected responses for a generated question-answer pair related to modality agreement tasks.

You will be provided with a detailed audio and video caption for a video clip.
Along with the caption, you will also be provided with the ground truth emotion label of the audio-visual clip.
Your task is to write a detailed audio-visual caption describing the emotional content of the video clip.
Keep the following points in mind while writing the caption:
1. The final caption should include both audio and visual elements.
2. Attribute the emotion only to the audio and visual cues in the captions which are relevant to the emotion label.
3. Do not ground the emotion description on any audio/visual cues which are not present in the provided captions.
4. Return your answer as a single paragraph.

Following is an example of how final audio-visual emotion caption should look like:
Example caption: "In the video, the opening scene shows a female character. She is looking directly at the other person, with her mouth slightly open, seemingly speaking or discussing a certain topic seriously. As time goes on, the character's expression becomes more excited and intense. The extent to which her mouth is open increases, possibly indicating that she is speaking loudly or arguing. In the following scene, the character's expression becomes more distorted, with a furrowed brow and downturned mouth, possibly indicating that her emotional state is escalating further. Based on these scenes, it can be inferred that the character in this video is likely experiencing a heated conversation or argument. In the audio, this character speaks with a strong tone, high volume, and fast pace. There are also continuous rhetorical questions with strong emotions. In the text, the subtitle reads: ""Is it useful to ask you? Are you ready to be a father? Luo Yiyang."" This sentence is likely spoken by the female character during the intense conversation or argument. Based on the changes in the female character's facial expressions from seriousness to excitement and further distorted expressions, as well as the description in the audio of the character's strong tone, high volume, and fast pace, we can infer that this sentence carries a sense of anger, dissatisfaction, or provocation. The female character may be questioning the other person's usefulness and readiness to be a father, expressing her discontent and anger."

If you think there are not enough audio-visual cues which support the emotion label, return a single word - "ERROR".

Now, write a detailed audio-visual caption for the following case
===
Video Caption: "{VIDEO_CAPTION}"
Audio Caption: "{AUDIO_CAPTION}"
Emotion Label: "{EMOTION_LABEL}"

Figure 36: **Audiovisual Caption Prompt** – used to combine audio and visual captions to create a combined audiovisual caption.

You will be provided with the ground truth description of a video capturing the emotional quotient of the video.
Your task is to evaluate a given model generation based by comparing it to the ground truth description.

You need to rate the model generation on a scale of 1 to 10 on the following criteria:
1. Audio-Visual Cue Overlap: Rate how well the mention of audio-visual events in the generation aligns with those in the ground truth. A higher score indicates a better match.
2. Emotion-label Consistency: Rate how accurately the predicted emotion from the model aligns with the emotion described in the ground truth. A higher score indicates better consistency.
3. Emotion-cue Association: Only focus on the model generation and rate how well the audio-visual cues are associated with the predicted emotion. Rate poorly if an emotion- irrelevant cue is mentioned in the generation. A higher score indicates a better association of audio-visual cues with emotion.
4. Hallucinated Cues: Rate the extent to which the model generations contains hallucinated or fabricated audio-visual cues that are not present in the ground truth. A higher score indicates fewer hallucinations and a lower score indicates more hallucinations.

Return your response in the following json format:
{"cue_overlap": int, "cue_overlap_reason": str, "emotion_consistency": int, "emotion_consistency_reason": str, "emotion_cue_association": int, "emotion_cue_association_reason": str, "hallucinated_cues": int, "hallucinated_cues_reason": str}

Ground Truth Description: *"{GT_DESCRIPTION}"*
Model generation: *"{MODEL_GENERATION}"*

Figure 37: **EMER Evaluation Prompt** – used to evaluate the model generations against the provided ground truths for the EMER dataset.

