# OpenReview forum: "AVERE: Improving Audiovisual Emotion Reasoning with Preference Optimization"
_ICLR.cc/2026/Conference — ICLR 2026 Poster_

### Official Review · Reviewer_D5KC · 2025-10-23

**Soundness:** 3
**Presentation:** 3
**Contribution:** 3
**Rating:** 8
**Confidence:** 4

**Summary:**

This paper focuses on emotion reasoning, addressing two key challenges: (1) Spurious associations between emotions and cues (reasoning errors), and (2) Hallucinations (perception errors). The authors first construct EmoReAIM, a benchmark designed to evaluate MLLMs on cue-emotion associations, hallucinations, and modality agreement. They then propose a framework, AVEm-DPO, which leverages DPO and text-prior debiasing techniques to enhance performance in emotion reasoning. Experimental results on three datasets demonstrate the effectiveness of their approach.

**Strengths:**

1.	This paper tackles a cutting-edge emotion reasoning task by proposing a novel benchmark and solution.

2.	The authors classify emotion reasoning errors into perception errors (hallucinations) and reasoning errors (spurious associations). This classification effectively covers most errors in emotion reasoning.

3.	The paper is well-written and well-motivated, with experimental results validating the effectiveness of the proposed method. It provides new insights into emotion reasoning.

**Weaknesses:**

1.	From my perspective, "Spurious AV cue associations" and "AV cue hallucinations" are essentially reasoning errors and perception errors, respectively, aligning with the errors in MLLMs. Thus, I suggest classifying errors primarily as reasoning and perception errors in the main text, followed by additional explanations from the "Spurious AV cue associations" and "AV cue hallucinations" perspectives. This would improve readability and attract broader interest from the MLLM research community.

2.	The paper provides a clear definition of "hallucinated cues" (i.e., descriptions containing clues that do not exist in the video). However, I have some concerns about "emotion-irrelevant cues". For example, in Figure 2 ("Emotion reasoning - basic"), the statement "The presence of light music in the background suggests a relaxed state" does relate to the person’s emotion state and should not be considered "emotion-irrelevant." Similarly, in Figure 1, "The dark green color of the background supports the negativity" could also be seen as a valid cue for inferring emotion. Overall, from my perspective, background music and color can be viewed as a useful clue for emotion reasoning and should not be dismissed as "emotion-irrelevant cues."

3.	The paper introduces two types of preference data, which is an interesting contribution. "Emotion-based Response Preference" aligns with my understanding of DPO. However, "Prompt-based Modality Preference" is less clear. Does it involve feeding different audiovisual inputs while expecting the same response? More examples would help clarify this type of DPO data.

**Questions:**

See weakness

---

> ### Author Response · Authors · 2025-11-21
> **Response to Weaknesses**
>
> We thank the reviewer for their time and valuable feedback. Please, find the responses to the raised concerns below:
>
> ### Classification to reasoning and perception errors
> ---
>
> We agree with the reviewer that *Spurious AV Cue-Emotion Associations* are reasoning errors made by the LLM to reason for the emotion present in the given input and *Emotion-relevant AV Cue Hallucinations* are perception errors that the LLM makes about perceiving the audio/visual cues that support a given emotion. We thank the reviewer for this suggestion. We have incorporated these changes to the **Abstract**, **Section 1** (paragraph 3), **Figure 1** (caption), **Section 3**  (paragraph 1), and **Section 3.1** (Emotion Reasoning-Stress Test).
>
> ### Emotion-irrelevant cues - Figures 1 and 2
> ---
>
> - **Emotion-irrelevant cues** are those audiovisual cues which do not contribute to the emotion of the person in the current scene. For example, in **Figure 2**, the presence of bicycles next to the man is irrelevant to his happiness.
> - **Figure 1** - We agree with the reviewer that the emotion-irrelevant cue in the illustration is poorly chosen. We have updated **Figure 1** in the manuscript.
> - **Figure 2** - We agree that the *presence of light music* is an emotion-relevant audio cue in the given sample. We have updated **Figure 2** in the manuscript with correct font color. However, since the question is about the *"man's speech"*, this option is still incorrect.
>
>
> ### Prompt-based Modality Preference
> ---
>
> Prompt-based Modality Preference builds explicit preference for getting the same response ($y_w$) with the correct modality inputs ($a_w$,$v_w$) over the rejected modality inputs ($a_l^{\text{PMP}}$, $v_l^{\text{PMP}}$). Here ($a_l^{\text{PMP}}$, $v_l^{\text{PMP}}$) depends on what the prompt is related to -- audio, visual or audiovisual. For example, consider the example given in **Figure 4** of the manuscript. If the prompt is vision-related e.g. *"How does the character's body language support their angry state?"*, then the correct modality inputs are the original audio and video from the input. The rejected modality inputs ($a_l^{\text{PMP}}$, $v_l^{\text{PMP}}$) for this case consist of the original audio ($a_w$) and a video with a different emotion ($v_l$) -- *i.e.*, the audio still remains the same because the prompt is video-related. This kind of input modality preference ensures that the model grounds its responses on the correct audio/visual cues instead of grounding them on something non-existent. Moreover, this also enforces that there is no cross-modal spurious interactions as we only *flip* the prompt-relevant modality input.

---

> > ### Comment · Reviewer_D5KC · 2025-11-24
> > **Response to Authors**
> >
> > Thanks for your reply. I'm willing to keep my score unchanged.

---

### Official Review · Reviewer_oa9K · 2025-10-27

**Soundness:** 4
**Presentation:** 4
**Contribution:** 3
**Rating:** 6
**Confidence:** 3

**Summary:**

The authors noticed that current models often misinterpret emotions by relying on irrelevant cues or hallucinated details, so they propose EmoReAlM, a benchmark for assessing audiovisual emotion reasoning and hallucination robustness in multimodal large language models (MLLMs), which contains 4,000 human-verified MCQA samples to evaluate emotion reasoning, modality alignment, and hallucination stress tests. Additionally, they also introduce AVEm-DPO, a Direct Preference Optimization approach that introduces prompt-based audiovisual preferences and text-prior debiasing to reduce hallucinations. Experiments show that AVEm-DPO significantly improves MLLM performance and robustness on emotion reasoning tasks compared with existing methods.

**Strengths:**

1. The paper is well-organized and easy to follow, with clear and informative tables and figures that effectively support the presentation.
2. To reduce hallucinations, the authors propose using Direct Preference Optimization (DPO). The method incorporates fine-grained, modality-level preferences based on the input text and reasoning about whether a response is hallucinatory or relevant to emotion prediction. Additionally, a text-prior debiasing strategy is introduced to mitigate hallucination effects. The overall approach is both reasonable and methodologically sound.
3. The experiments and ablation studies are comprehensive, and the presentation of results is clear and easy to follow.

**Weaknesses:**

1. What is the motivation to use LLMs for visual and audio emotion prediction? It is challenging for LLMs to accurately infer emotions based solely on captions, even for advanced models such as GPT-4o. Moreover, even when an LLM’s prediction matches the ground truth, it does not necessarily imply that the emotional trigger or the reasoning process behind the prediction is correct.

2. Is there any analysis on the individual roles of the visual and audio modalities? For example, which modality provides the essential cues for the final prediction, and do we need both modalities for effective emotion reasoning?

3. In data creation, captions play a crucial role in translating visual information into text. Was any sampling strategy applied during this process? How do the authors ensure that the essential frames or emotional triggers are preserved during caption generation?

**Questions:**

None

---

> ### Author Response · Authors · 2025-11-21
> **Response to Weaknesses (1/2)**
>
> We thank the reviewer for their time and feedback to our work. We request the reviewer to find the responses to the points mentioned in the weakness below:
>
> ### Motivation for using LLMs for emotion understanding
> ---
>
> - As mentioned in the introduction (second paragraph) of the original manuscript, traditional emotion recognition techniques often lack contextual understanding of audiovisual cues. Natural language explanations are necessary for critical use cases, e.g., health and education where human operators prefer nuanced explanations rather than simple labels from black box models. Hence, similar to prior work [R1][R2], we use multimodal LLMs for emotion recognition.
> - We agree with the last statement in the first weakness of the reviewer regarding correctness of the trigger or the reason for emotion prediction within LLMs. Our work directly addresses this problem. Instead of emotion prediction using traditional benchmarks which only test the model's ability to predict the correct emotion label, the proposed EmoReAlM benchmark contains various tasks which test the emotion reasoning capabilities of multimodal LLMs. The following tasks require an understanding of emotional triggers and associations of audiovisual cues with emotions.
>     - Emotion Reasoning - Basic: tests the basic reasoning abilities of multimodal LLMs for identifying and associating correct audiovisual cues to emotions.
>     - Emotion Reasoning - Stress Test: tests spurious audiovisual cue and emotion associations, along with testing audiovisual hallucinations in LLMs for explaining emotions.
>
>
> ### Individual roles of audio and visual modalities
> ---
>
> To show the effect of using individual modalities for emotion prediction, we pass the audio and video from the RAVDESS dataset individually to different models and report emotion recognition performance in **Table R1**. We can clearly observe that for almost all models, the performance of the video-only model  is higher than that of the audio-only ones. This underlines the importance of the visual modality for multimodal categorical emotion recognition. We have added this discussion in **Appendix E.10 Table 19** of the manuscript.
>
> **Table R1** - *Roles of individual modalities for emotion prediction*
> | Model | AV (UAR) | AV (WAR) | Only-V (UAR) | Only-V (WAR) | Only-A (UAR) | Only-A (WAR) |
> |---|---|---|---|---|---|---|
> | VideoLLaMA 2 | 41.81 | 31.62 | 36.12 | 32.41 | 30.44 | 27.56 |
> | Qwen 2.5 Omni | 32.88 | 28.05 | 29.38 | 27.67 | 28.56 | 25.55 |
> | Our base | 53.59 | 53.01 | 41.27 | 40.98 | 38.18 | 37.74 |
> | + AVEm-DPO | 58.66 | 55.48 | 46.13 | 46.31 | 44.05 | 39.67 |
> | EmotionLLaMA | 52.59 | 48.12 | 41.27 | 39.56 | 38.57 | 37.27 |
> | + AVEm-DPO | 56.21 | 51.03 | 46.10 | 43.09 | 40.54 | 37.04 |
>
> [R1] Zebang Cheng, Zhi-Qi Cheng, Jun-Yan He, Jingdong Sun, Kai Wang, Yuxiang Lin, Zheng Lian, Xiaojiang Peng, and Alexander G. Hauptmann. 2024. Emotion-LLaMA: multimodal emotion recognition and reasoning with instruction tuning. In Proceedings of the 38th International Conference on Neural Information Processing Systems (NeurIPS '24), Vol. 37. Curran Associates Inc., Red Hook, NY, USA, Article 3518, 110805–110853.
>
> [R2] Zhiyuan Han, Beier Zhu, Yanlong Xu, Peipei Song, and Xun Yang. 2025. Benchmarking and Bridging Emotion Conflicts for Multimodal Emotion Reasoning. In Proceedings of the 33rd ACM International Conference on Multimedia (MM '25). Association for Computing Machinery, New York, NY, USA, 5528–5537. https://doi.org/10.1145/3746027.3754856

---

> ### Author Response · Authors · 2025-11-21
> **Response to Weaknesses (2/2)**
>
> ### Data sampling for visual captioning
> ---
>
> As mentioned in **Appendix B.1**, we use 8 uniformly sampled frames for visual captioning through GPT-4o. This choice is empirically motivated and we did the following experiment to justify our choice. For the EMER [R3] dataset, we extract the visual caption from GPT-4o by providing 1, 2, 4, 8, and 16 frames uniformly sampled from the videos. EMER dataset (mean duration - 3.78s) is similar to DFEW (mean duration - 3.42s) in terms of the duration of the audiovisual content. We report text similarity between the generated caption and the ground truth labels using SentenceBERT [R4] cosine similarity and BERT Score [R5]. We can clearly note that using 8 frames leads to significantly higher scores compared to using less frames. Using 16 frames does not provide additional boost, in return for the significantly higher cost. Also, using 8 frames for the DFEW dataset with average duration of 3.42 seconds leads to ~2FPS sampling rate on an average. This discussion has been added to **Appendix B.4 Table 8** of the manuscript.
>
> **Table R2** - *Effect of Frame Sampling on Visual Captioning*
> | # frames | S-BERT Sim. | BERT Score (Prec.) | BERT Score (Rec.) | BERT Score (F1) |
> |---|---|---|---|---|
> | 1 | 0.646 | 0.851 | 0.853 | 0.852 |
> | 2 | 0.660 | 0.851 | 0.856 | 0.853 |
> | 4 | 0.676 | 0.851 | 0.857 | 0.854 |
> | 8 | 0.689 | 0.858 | 0.861 | 0.860 |
> | 16 | 0.688 | 0.858 | 0.862 | 0.860 |
>
> [R3] Lian, Zheng, Haiyang Sun, Licai Sun, Hao Gu, Zhuofan Wen, Siyuan Zhang, Shun Chen et al. "Explainable multimodal emotion recognition." arXiv preprint arXiv:2306.15401 (2023).
>
> [R4] Reimers, Nils, and Iryna Gurevych. ‘Sentence-BERT: Sentence Embeddings Using Siamese BERT-Networks’. In Proceedings of the 2019 Conference on Empirical Methods in Natural Language Processing. Association for Computational Linguistics, 11 2019. https://arxiv.org/abs/1908.10084.
>
> [R5] Zhang, Tianyi, Varsha Kishore, Felix Wu, Kilian Q. Weinberger, and Yoav Artzi. "BERTScore: Evaluating Text Generation with BERT." In International Conference on Learning Representations.

---

> > ### Comment · Reviewer_oa9K · 2025-11-24
> >
> > Thank you for the comments—they are helpful in improving my understanding of the paper. However, I still have some concerns. Specifically, how is the ground-truth reasoning obtained, like for emotion reasoning, how to define the correct visual and audio clue, and how is the audio–frame alignment performed, given that the video frames are sampled and may not be temporally correlated with the audio signal? Moreover, according to Table R2, a single-frame model appears to achieve strong performance, suggesting that temporal information may contribute only marginally. Could the authors clarify this?

---

> > > ### Author Response · Authors · 2025-11-25
> > >
> > > We thank the reviewer for checking our paper and our reply again.
> > >
> > > ### Regarding obtaining ground truth reasoning
> > > ---
> > > As mentioned in **Section 3.2 - Figure 3** (detailed in **Appendix B.1**), we extract detailed audio and visual captions for the given input using prompts from GPT-4o (using prompts in **Figures 19 and 20**). To automatically ensure that the generated visual and audio captions indeed are correct, we predict the emotion from the generated audio and visual captions using GPT-4o (using prompts in **Figures 21 and 22**). If the predicted emotion in this step is indeed the ground truth (GT) emotion (we have the GT emotion because we extract our dataset from existing emotion recognition datasets), then this is a good signal that the generated audio/visual caption contains correct clues to determine the GT emotion.  The rest of the pipeline then involved using the captions, which led to correct emotion predictions and rephrasing them into audio/visual reasoning questions using GPT-4o (using prompts in **Figures 23 and 24**). This is a (cyclical) self-verified way of generating QA for emotion reasoning. To further ensure that the emotion reasoning is correct, we perform human verification on the generated reasoning for EmoReAlM Benchmark as mentioned in **Section 3.3** and **Appendix B.2**.
> > >
> > > ### Regarding audio-frame alignment
> > > ---
> > > - **Data generation** - *Emotion Reasoning (Basic and Stress Tests)* tasks in EmoReAlM Benchmark are for individual modalities and do not require any audio-frame alignment. For the *Modality Agreement* task, since the task is about whether the emotion inferred from the audio and video modalities (**overall**) is in agreement or not, audio-frame alignment is not a concern. Hence, we extract the captions for the entire audio and video and use them for downstream processing as mentioned in **Section 3.2** (and **Appendix B.1**).
> > > - **Model architecture** - As mentioned in **Section 5 (Reference models)** (and **Appendix D.2**), our architecture is similar to EmotionLLaMA [R1].  We extract the audio and video features from respective encoders and concatenate the tokens before feeding them into the language model.
> > >
> > > We acknowledge that our benchmark and model have limitations for fine-grained temporal reasoning, which should be addressed in future work.
> > >
> > > ### Single-frame contribution to visual information
> > > ---
> > > We agree with the reviewer that for even a single frame, the quality of visual captions is high. However, to show the effect of using more frames, **Table R3** has the generated caption and ground truth reasoning similarity scores on the long-duration clips from EMER (>=5 seconds - 83/332 videos). This shows that there is indeed temporal information in the video clips, and using more frames helps in generating better video captions.
> > >
> > >
> > > **Table R3** - *Effect of Frame Sampling on Visual Captioning on Long Videos(>=5 seconds)*
> > > | # frames | S-BERT Sim. | BERT Score (Prec.) | BERT Score (Rec.) | BERT Score (F1) |
> > > |---|---|---|---|---|
> > > | 1 | 0.598 | 0.819 | 0.814 | 0.816 |
> > > | 2 | 0.616 | 0.832 | 0.828 | 0.830 |
> > > | 4 | 0.645 | 0.847 | 0.843 | 0.845 |
> > > | 8 | 0.693 | 0.860 | 0.858 | 0.859 |
> > > | 16 | 0.698 | 0.864 | 0.859 | 0.861 |
> > >
> > > [R1] Zebang Cheng, Zhi-Qi Cheng, Jun-Yan He, Jingdong Sun, Kai Wang, Yuxiang Lin, Zheng Lian, Xiaojiang Peng, and Alexander G. Hauptmann. 2024. Emotion-LLaMA: multimodal emotion recognition and reasoning with instruction tuning. In Proceedings of the 38th International Conference on Neural Information Processing Systems (NeurIPS '24), Vol. 37. Curran Associates Inc., Red Hook, NY, USA, Article 3518, 110805–110853.

---

> > > > ### Comment · Reviewer_oa9K · 2025-11-25
> > > >
> > > > Thank you for the demonstration. I don’t have any further questions, and I’m willing to keep my current rate.

---

### Official Review · Reviewer_Hdfz · 2025-10-31

**Soundness:** 2
**Presentation:** 3
**Contribution:** 3
**Rating:** 4
**Confidence:** 4

**Summary:**

This paper introduce a benchmark designed to evaluate MLLMs for cue–emotion associations, hallucinations and modality agreement and propose a preference optimization technique that aligns model responses with both audiovisual inputs and emotion-centric queries.

**Strengths:**

- A comprehensive suite of 4,000 human-verified multiple-choice questions (MCQs) across 2,649 unique videos, designed to evaluate three critical aspects of emotion reasoning.
- A multimodal direct preference optimization (DPO) method to align MLLMs with both audiovisual inputs and emotion-centric queries.
- Demonstrates that AVEm-DPO outperforms baselines by 6–19% in zero-shot settings across existing benchmarks and EmoReAlM, with qualitative and user studies confirming reduced hallucinations and spurious associations.

**Weaknesses:**

- EmoReAlM is derived exclusively from the DFEW dataset, which may limit generalizability to videos with different cultural contexts, demographics, or emotion types
- AVEm-DPO’s training data is generated automatically via Gemini 2.5 (without human verification). While the authors report performance gains, unvalidated preference pairs may introduce hidden biases

**Questions:**

See weakness

---

> ### Author Response · Authors · 2025-11-21
> **Response to Weaknesses**
>
> We thank the reviewer for their time and valuable feedback. The responses to the reviewer's concerns are given below.
>
> ### EmoReAlM extracted only from DFEW
> ---
>
> We agree that extracting EmoReAlM from the DFEW benchmark can be a point of concern, however, DFEW is a large and diverse dataset (>16k videos) covering audiovisual samples from varied languages/cultures and emotions. Following is the distribution of languages and emotions present in EmoReAlM benchmark.
>
> **Table R1** - *Language Distribution*
> | Language | Count | % of Total |
> |----------|-------|------------|
> | English | 1163 | 43.89% |
> | Chinese | 285 | 10.75% |
> | Japanese | 151 | 5.70% |
> | Korean | 116 | 4.38% |
> | French | 50 | 1.89% |
> | Hindi | 24 | 0.91% |
> | Italian | 23 | 0.87% |
> | Spanish | 21 | 0.79% |
> | Russian | 17 | 0.64% |
> | Nynorsk | 10 | 0.38% |
> | Turkish | 9 | 0.34% |
> | Hungarian | 7 | 0.26% |
> | Swedish | 6 | 0.23% |
> | German | 6 | 0.23% |
> | Polish | 5 | 0.19% |
> | Others | 30 | 1.13% |
>
> **Table R2** - *Emotion Distribution (also in Appendix B.3-Fig.7 (right))*
> | Emotion | Count | % of Total |
> |---------|-------|------------|
> | Happiness | 854 | 21.35% |
> | Anger | 729 | 18.22% |
> | Sadness | 692 | 17.30% |
> | Surprise | 317 | 7.92% |
> | Fear | 257 | 6.42% |
> | Disgust | 11 | 0.27% |
> | Neutral | 1140 | 28.50% |
>
> As shown in **Appendix B.3 - Fig. 7 (right)**, the emotion distribution is similar to the original data distribution for DFEW. We obtain the language distribution in **Table R1** using Whisper to detect the language present in the samples which contain speech.
>
> We have incorporated the language distribution statistic for EmoReAlM in **Appendix B.3 Fig. 8** of the manuscript.
>
> ### Training data generated from Gemini
> ---
>
> To validate the automatically generated training data from Gemini, we have performed a human verification study on Prolific for 1000 samples with 90 participants. For each sample, we ask 3 annotators to annotate the chosen, rejected (emotion relevant) and rejected (video relevant) responses. The following table reports the number of samples in which majority of annotators found that the response is correct (*i.e.*, either correct, video-relevant, or emotion-relevant). Additionally, we also report the number of samples in which one or more annotators felt that the provided responses are correct. We can clearly observe that on this random subset of the preference dataset, more than 90% of the chosen responses are correct showing that the automatically generated data is indeed valid. This underlines the validity of the our preference data generation pipeline. We have added the above discussion as **Section 5.3** of the manuscript.
>
> **Table R3** - *Human Verification of Generated Preference Data*
> | Response Type | # Total | # Majority Correct | # Correct (1 or more) |
> |---|---|---|---|
> | Chosen ($y_w$) | 1000 | 912 | 967 |
> | Rejected - Video Relevant ($y_l^{vr}$) | 1000 | 895 | 923 |
> | Rejected - Emotion Relevant ($y_l^{er}$) | 1000 | 856 | 912 |

---

> > ### Comment · Reviewer_Hdfz · 2025-11-24
> >
> > Thanks for the authors’ response and I am willing to increase my score to 6.

---

### Official Review · Reviewer_poWb · 2025-11-01

**Soundness:** 3
**Presentation:** 3
**Contribution:** 3
**Rating:** 6
**Confidence:** 4

**Summary:**

This paper investigates audiovisual emotion reasoning in multimodal large language models (MLLMs). It identifies two key problems: spurious associations between emotion labels and irrelevant audiovisual cues, and hallucination of cues (models inventing non-present audio/visual evidence). To address evaluation gaps, the authors introduce EmoReAlM, a large benchmark with tasks for emotion reasoning (basic and stress-test) and modality agreement, including human verification and statistics drawn from sources such as DFEW. For optimization they propose AVEm-DPO, a preference-optimization approach that combines explicit preferences (prompt-based modality preference and emotion-based response preference) with a text-prior debiasing (TPD) regularizer to reduce reliance on textual priors. Experiments across many baselines and ablations show substantial gains: AVEm-DPO improves zero-shot performance and reduces hallucination and spurious associations (reported improvements of ~6–19% in several settings, and large gains in user-evaluation metrics). The paper provides extensive benchmark details, human verification procedures, and ablative analyses demonstrating which components contribute most to the improvements.

**Strengths:**

- EmoReAlM benchmark: A comprehensive, human-verified benchmark for audiovisual emotion understanding that tests (a) cue-emotion associations, (b) modality agreement, and © robust stress tests designed to reveal spurious associations and hallucinations. The benchmark includes balanced tasks, adversarial cases, and metrics for spurious associations, modality agreement, and hallucination.
- AVEm-DPO optimization framework: A novel preference optimization method tailored to audiovisual emotion reasoning. It integrates prompt-based modality preference (PMP) and emotion-based response preference (ERP) with text-prior debiasing (TPD) to steer MLLMs toward using actual audiovisual cues and away from text priors or hallucinated cues. The method is evaluated against several baselines and ablations.
- Extensive evaluation and analysis: Thorough quantitative and qualitative experiments (including human evaluation) that show large improvements in emotion attribution, reduced hallucination, and better modality alignment. The paper includes ablation studies, sensitivity analyses, class-wise results, and adversarial modality tests that clarify which components are most effective.

**Weaknesses:**

- Reliance on proprietary or large LLM tooling for some steps: The paper mentions using GPT-5 to polish text and using LLMs for annotation/evaluation. This reliance can raise reproducibility concerns if those tools or their prompts are not fully disclosed; it may also bias dataset construction and evaluation unless careful controls are provided.
- Potential dataset and evaluation biases: Although the benchmark is human-verified, the document suggests many generated QA items and uses subtitled non-English videos with English subtitles. This raises questions about cultural, linguistic, or subtitle-transcription biases influencing emotion labels and whether the benchmark fully generalizes across languages, contexts, and production styles.
- Limited discussion of failure modes and external validity: While many stress tests and ablations are provided, the paper could better characterize remaining failure cases (e.g., what kinds of emotions or multimodal interactions still confuse the model), and whether AVEm-DPO overfits to particular datasets or annotation styles. Also, practical deployment implications (latency, compute cost, or safety concerns when misclassifying emotions) are not deeply discussed.

**Questions:**

SEE WEAKNESS

---

> ### Author Response · Authors · 2025-11-21
> **Response to Weaknesses (1/2)**
>
> We thank the reviewer for their time and valuable feedback. Please, find the responses to the raised concerns below:
>
> ### Details about usage of proprietary LLMs
> ---
>
> We acknowledge the reviewer's concern regarding the usage of proprietary LLMs. All prompts used in construction of the proposed dataset with details about their usage are available in the following sections of the Appendix:
> - Benchmark creation - **Appendix B.1**
>     - Audio and visual captioning - **Fig. 19-20**
>     - Emotion prediction from audio and video captions - **Fig. 21-22**
>     - QA Generation - **Fig. 23-31**
>     - Post-processing - **Fig. 32**
> - Preference data construction - **Appendix C.2** **(Fig. 33-36)**
> - Evaluation using GPT - **Appendix D.1 (Fig. 37)**
>
> Additionally, to reduce the risk of LLM data generation bias, we use a GPT-4o with human verification for benchmark creation. However, the preference data was generated by Gemini-2.5 without any further vertification (as mentioned in **Sec. 4.3** & **Appendix C.2**).
>
> We would also like to ask the reviewer to further clarify the following comment --
> ```
> it may also bias dataset construction and evaluation unless careful controls are provided
> ```
>
> ### Potential dataset and evaluation biases
> ---
> - **Dataset biases**
>     - *"biases influencing emotion labels"* - As described in Sec. 3.2, our benchmark construction pipeline utilizes emotion labels from an existing audiovisual emotion recognition dataset - DFEW [R1]. DFEW is a large-scale human-annotated dataset widely used as a benchmark for multimodal emotion recognition [R2-R3]. Hence, our human annotators only have to verify the generated emotion reasoning QAs, by paying attention to the audiovisual cues.
>     - Moreover, past work on emotion perception (perceiving an expression of emotion) [R4] shows low variation across cultures, further supporting the validity of our human verification pipeline.
> - **Evaluation biases**
>     - We would like to clarify that all model training and evaluations are performed only using audiovisual inputs without any text-subtitle. Hence, subtiltes will not bias model training or evaluation.
>
> [R1] Jiang, Xingxun, Yuan Zong, Wenming Zheng, Chuangao Tang, Wanchuang Xia, Cheng Lu, and Jiateng Liu. "Dfew: A large-scale database for recognizing dynamic facial expressions in the wild." In Proceedings of the 28th ACM international conference on multimedia, pp. 2881-2889. 2020.
>
> [R2] Cheng, Zebang, Zhi-Qi Cheng, Jun-Yan He, Kai Wang, Yuxiang Lin, Zheng Lian, Xiaojiang Peng, and Alexander Hauptmann. "Emotion-llama: Multimodal emotion recognition and reasoning with instruction tuning." Advances in Neural Information Processing Systems 37 (2024): 110805-110853.
>
> [R3] Zhiyuan Han, Beier Zhu, Yanlong Xu, Peipei Song, and Xun Yang. 2025. Benchmarking and Bridging Emotion Conflicts for Multimodal Emotion Reasoning. In Proceedings of the 33rd ACM International Conference on Multimedia (MM '25). Association for Computing Machinery. https://doi.org/10.1145/3746027.3754856
>
> [R4] Fang, Xia, Gerben A. van Kleef, and Disa A. Sauter. "Revisiting cultural differences in emotion perception between easterners and westerners: Chinese perceivers are accurate, but see additional non-intended emotions in negative facial expressions." Journal of Experimental Social Psychology 82 (2019): 152-159.

---

> ### Author Response · Authors · 2025-11-21
> **Response to Weaknesses (2/2)**
>
> ### Limitations and Practical Considerations
> ---
>
> - **Failure modes**
>     - **Appendix E.2 Table 14** shows the performance of our approach on different subtasks of the Emotion Reasoning-Stress Tests of EmoReAlM. We can observe that the model performance on spurious associations of audio cues with emotion labels is significantly worse than the other sub-tasks. This shows that there are still underlying spurious audio cue-emotion associations in the model which is a potential limitation.
>     - **Appendix E.3 Table 15** shows the results for emotion recognition on different emotion labels of the DFEW dataset. Although the proposed method outperforms several baselines on the *disgust* emotion class, there is still a lot of room for improvement. We attribute this performance gap to the limited number of samples in the existing benchmarks for *disgust* (an ambiguous expression [R5]) and consequently the limited number in the constructed preference dataset.
> - **Compute and Latency** - Since we propose a post-training technique without adding any parameters to the original model, there is no additional compute and latency overhead for a model trained through AVEm-DPO during evaluation/inference.
> - **Safety concerns** - We thank the reviewer to point this out. Similar to past works on emotion recognition and mentioned in **Appendix E.3 (Table 15)**, our method works slightly worse on the *disgust* emotion compared to the other emotions. We acknowledge that the proposed work can lead to potential safety concerns in such cases.
>
> We have added a **Limitations and Future Works** section (**Section 6**) to the manuscript to incorporate some of the aforementioned points. Safety concerns have been added in the **Ethics Statement** of the manuscript.
>
>
> [R5] Emalie Hendel et al. "Exploration of visual factors in the disgust-anger confusion: the importance of the mouth." Cognition and Emotion 37.4 (2023): 835-851.

---

### Author Response · Authors · 2025-12-04
**Summary of the reviews and responses**

We thank all the reviewers for their feedback and comments regarding our paper. We are grateful to the reviewers for giving a positive final rating (>=6) to our paper (Reviewer `Hdfz` increased their score after our response, as mentioned in their [comment in the discussion thread](https://openreview.net/forum?id=td682AAuPr&noteId=UqBBcO0TMv)). Specifically, the reviewers liked our **comprehensive emotion reasoning benchmark -- EmoReAlM** (Reviewers `poWb`, `Hdfz`, `D5KC`), **preference optimization technique -- AVEm-DPO** (Reviewers `poWb`, `Hdfz`, `oa9K`, `D5KC`), **experimental analysis and evaluation** (Reviewers `poWb`, `Hdfz`, `oa9K`), and **paper-writing and organization** (Reviewers `D5KC`, `oa9K`).

To the best of our efforts, we addressed all the reviewer comments and questions regarding the following areas:
- Limitations and practical considerations (Reviewer `poWb`) - Added a **Limitations and Future Works** section **(Section 6)**.
- Cultural diversity of EmoReAlM (Reviewer `Hdfz`) - Added cultural (language) distribution plot in **Appendix B.3 Fig. 8**.
- Training data validation (Reviewer `Hdfz`) - Added human verification results for the training data in **Section 5.3**.
- Individual roles of audio and visual modalities (Reviewer `oa9K`) - Added discussion and results in **Appendix E.10 Table 19**.
- Details about visual captioning for data annotation (Reviewer `oa9K`) - Added appropriate discussion in **Appendix B.4 Table 8**.
- Classification of errors (Reviewer `D5KC`) - Updated the definitions in the appropriate sections (**Abstract**, **Section 1** (paragraph 3), **Figure 1** (caption), **Section 3**  (paragraph 1), and **Section 3.1** (Emotion Reasoning-Stress Test).)

We thank the reviewers for acknowledging our responses to their reviews and for engaging in discussion to improve our work. We are optimistic that our work will push the boundaries of emotion reasoning and social AI in multimodal LLMs.

---

### Meta-Review · Area_Chair_irff · 2026-01-07

**Summary:**

**Strengths**:

1. Introduces EmoReAlM, a large-scale, human-verified benchmark with balanced tasks and robust stress tests for hallucinations, spurious associations, and modality agreement
(poWb, Hdfz, D5KC).

2. Proposes AVEm-DPO, a principled and tailored preference optimization framework for audiovisual emotion reasoning
(poWb, Hdfz, oa9K).

3. Extensive experimental validation, including ablations and human/user studies, showing consistent improvements in grounding and reduced hallucination
(poWb, Hdfz, oa9K, D5KC).

4. Clear problem formulation and presentation; the paper is well-organized and easy to follow with informative figures and tables
(oa9K, D5KC).

**Weaknesses**:

1. Generalizability concerns due to reliance on a single source dataset (DFEW), generated QA items, and subtitle-based supervision
(poWb, Hdfz).

2. Reproducibility risks stemming from the use of proprietary or large LLMs (e.g., GPT-5, Gemini 2.5) for data generation, polishing, and evaluation
(poWb, Hdfz).

3. Conceptual ambiguities in cue definitions, particularly the treatment of “emotion-irrelevant cues” (e.g., background music or color)
(D5KC).

4. Limited analysis of individual modality contributions (audio vs. visual) and the motivation for LLM-based emotion prediction
(oa9K).

5. Insufficient clarification of prompt-based modality preference construction and limited discussion of failure modes and deployment considerations
(D5KC, poWb).

**Reviewer Concerns:**

Most of the reviewers’ concerns have been adequately addressed, and no crucial issues remain.

**Reviewer Scores:**

- Reviewer poWb: 6 -> 6
- Reviewer Hdfz: 4 -> 6
- Reviewer oa9K: 6 -> 6
- Reviewer D5KC: 8 -> 8

---

### Decision · Program_Chairs · 2026-01-26

Accept (Poster)